# A systematic review of the prediction of hospital length of stay: Towards a unified framework

**Kieran Stone** [1]*, **Reyer Zwiggelaar** [1], **Phil Jones**[2‡], **Neil Mac Parthaláin**[1]

**1** Department of Computer Science, Aberystwyth University, Ceredigion, SY23 3DB, Wales, United Kingdom,
**2** Bronglais District General Hospital, Aberystwyth, Ceredigion, SY23 1ER, Wales, United Kingdom

‡Unavailable
* kis12@aber.ac.uk

**Data Availability Statement:** The paper is a Systematic Review or Survey of the Hospital Length of Stay prediction field. As such there is no direct data that has been created from this study

## Abstract

Hospital length of stay of patients is a crucial factor for the effective planning and management of hospital resources. There is considerable interest in predicting the LoS of patients in order to improve patient care, control hospital costs and increase service efficiency. This paper presents an extensive review of the literature, examining the approaches employed for the prediction of LoS in terms of their merits and shortcomings. In order to address some of these problems, a unified framework is proposed to better generalise the approaches that are being used to predict length of stay. This includes the investigation of the types of routinely collected data used in the problem as well as recommendations to ensure robust and meaningful knowledge modelling. This unified common framework enables the direct comparison of results between length of stay prediction approaches and will ensure that such approaches can be used across several hospital environments. A literature search was conducted in PubMed, Google Scholar and Web of Science from 1970 until 2019 to identify LoS surveys which review the literature. 32 Surveys were identified, from these 32 surveys, 220 papers were manually identified to be relevant to LoS prediction. After removing duplicates, and exploring the reference list of studies included for review, 93 studies remained. Despite the continuing efforts to predict and reduce the LoS of patients, current research in this domain remains ad-hoc; as such, the model tuning and data preprocessing steps are too specific and result in a large proportion of the current prediction mechanisms being restricted to the hospital that they were employed in. Adopting a unified framework for the prediction of LoS could yield a more reliable estimate of the LoS as a unified framework enables the direct comparison of length of stay methods. Additional research is also required to explore novel methods such as fuzzy systems which could build upon the success of current models as well as further exploration of black-box approaches and model interpretability.

however data that has been used in each study that is considered within the review is described within this manuscript.

**Funding:** Kieran Stone would like to acknowledge the financial support for this research through Knowledge Economy Skills Scholarship (KESS 2). It is part funded by the Welsh Government's European Social Fund (ESF) convergence programme for West Wales and the Valleys. WEFO (Welsh European Funding Office) contract number: C80815. The funders had no role in study design, data collection and analysis, decision to publish, or preparation of the manuscript.

**Competing interests:** The authors have declared that no competing interests exist.

## Author summary

Hospital length of stay (LoS) is the number of days that an in-patient will remain in hospital. LoS has long been used as a measure for hospitals so that they can better improve patient care, reduce overall costs, and appropriately allocate resources according to staff and patient needs. It can also give an indication of hospital care unit efficiency and patient flow. There is of course much considerable variability There is considerable variability amongst patient LoS for different patient diagnoses. The LoS for the same diagnosis may vary from 2 to 50+ days between patients. This variation can be due to several factors such as a patient's characteristics, social circumstances, or treatment complexity. This paper explores the makeshift nature of the current LoS prediction approaches and highlights the need for a unified framework to be adopted which could yield a more reliable estimate of LoS. This framework would enable the performance of several LoS prediction approaches to be directly compared and could be used across several hospital environments. Expanding the influence of these models that are generated as part of a unified framework would ensure that the prediction approaches in place are suitably robust.

## 1 Hospital length of stay

In order to ensure optimal levels of care, healthcare systems have begun to place increasing emphasis on effective resource management and forecasting to reduce the associated cost and improve patient care [1]. The primary focus of hospital managers is to establish appropriate healthcare planning by allocating facilities and necessary human resources required for efficient hospital operation in accordance with patient needs. Several approaches have been developed to predict admissions, patient bed needs and overall bed utilisation for healthcare systems. The most important aspect of such approaches is a reliable prediction of how long a patient who is admitted to hospital will stay and an understanding of the factors which have a strong influence on patient length of stay.

Patient hospital length of stay (LoS) can be defined as the number of days that an in-patient will remain in hospital during a single admission event [2]. As well as being one of the major indicators for the consumption of hospital resources, LoS can also provide an enhanced understanding of the flow of patients through hospital care units and environments, which is a significant factor in the evaluation of operational functions of various care systems. LoS is often considered a metric which can be used to identify resource utilisation, cost and severity of illness [3] [4]. Previous work has sought to group patients by their respective medical condition (s), which assumes that each disease, condition or procedure is associated with a predefined, recommended LoS [5]. However, LoS is a much more complex concept, which can be affected by a multitude of different (sometimes competing) factors including (but not limited to): a patient's characteristics, presenting complaint, complications and discharge planning, as well as treatment complexity, all of which are likely to extend the original target LoS. As such, a model that has the ability to reliably predict patient LoS during a single visit event, could be an effective method for healthcare services to action preventative measures in order to avoid the extension of LoS. The majority of patients would prefer to be cared for in the comfort of their own homes, if suitable care can be provided, particularly in terms of palliative care [6]. Additionally, there is potential harm for patients who remain in hospital longer than required for active care. In hospital, falls, hospital acquired infections and medication errors which occur in patients who are fit for discharge need to be avoided as they will prolong a patient's LoS [7]. Proactively managing discharge from as early in the admission as possible and reducing length of stay would help

to protect patients and hospitals from such complications [8]. From the point of view of the healthcare provider, reducing the LoS of patients is desirable for two reasons. The first of which focuses on the needs of the patient and adjusting the level of care received specifically to meet the needs of each patient [9]. The second relates to the overall management and planning of healthcare resource and aims to reduce LoS by reducing the volume of resource that is invested in any single patient so that the resource can be shared with others [10].

The scope of this systematic review is to evaluate recent developments that are related to the domain of LoS prediction and draw conclusions about the state-of-the-art approaches that are used for LoS prediction. Various approaches that have been employed to predict LoS are reviewed, along with their relative merits and shortcomings. This work highlights the many challenges of predicting LoS as well as the current gaps in the literature and how they might be addressed. The remainder of the paper is structured as follows: In Section 2, the study review process is described. In Section 3, the different approaches that have been employed to model and predict LoS are presented in terms of their suitability in assessing LoS. Section 4 documents the types of data that have been used in LoS prediction are discussed as well as the features in the data that are commonly considered to have an influence on LoS. In Section 5, an appraisal of the current state-of-the-art is provided. In Section 6, the gaps in the literature and the potential areas for improvement are discussed along with a framework for addressing the current shortcomings. Finally some conclusions are drawn and topics for further exploration are highlighted.

## 2 Methods

This survey makes use of a rapid evidence assessment (REA) methodology which is structured using the Preferred Reporting Items for Systematic Reviews and Meta Analyses (PRISMA) checklist [11]. REA uses similar principles to that of a systematic review but makes concessions to the depth of the process in order to address the key issues that are important to the topic under investigation. In this survey, the search was restricted to papers written between the 1970s until 2019, written in the English language and selected from exploring major electronic databases. In what follows, the search strategy, eligibility criteria, data extraction process and quality assessment are described. Please note: A protocol does not exist for this systematic review.

### 2.1 Search strategy

A literature search was conducted in PubMed, Google Scholar and Web of Science from 1970 until 2021 to identify LoS surveys which review the literature. The term 'surveys' represents papers which encapsulate and review a large number of papers within the field, these are not exclusively limited to systematic reviews. These sources were chosen as they have the broadest range of relevant content to the review topic. The search terms and synonyms were used to locate surveys that were relevant to LoS prediction are described in Table 1. The search

**Table 1. Search terms.**

| Search Term |
| --- |
| "Duration of stay prediction" |
| "Hospital stay prediction" |
| "Length of stay" or "Length-of-stay" |
| "Length of Hospital Stay" |
| "Predict" or "Predictive" or "Prediction" or "Predictor" |
| "Bed occupancy modelling prediction" |

comprised the fields "title"; "abstracts"; and "keywords". 32 Surveys were identified, from these 32 surveys, 220 papers were manually identified to be relevant to LoS prediction. After removing duplicates, and exploring the reference list of studies included for review, 93 studies remained. The selection process followed the PRISMA checklist and is shown in Fig 1.

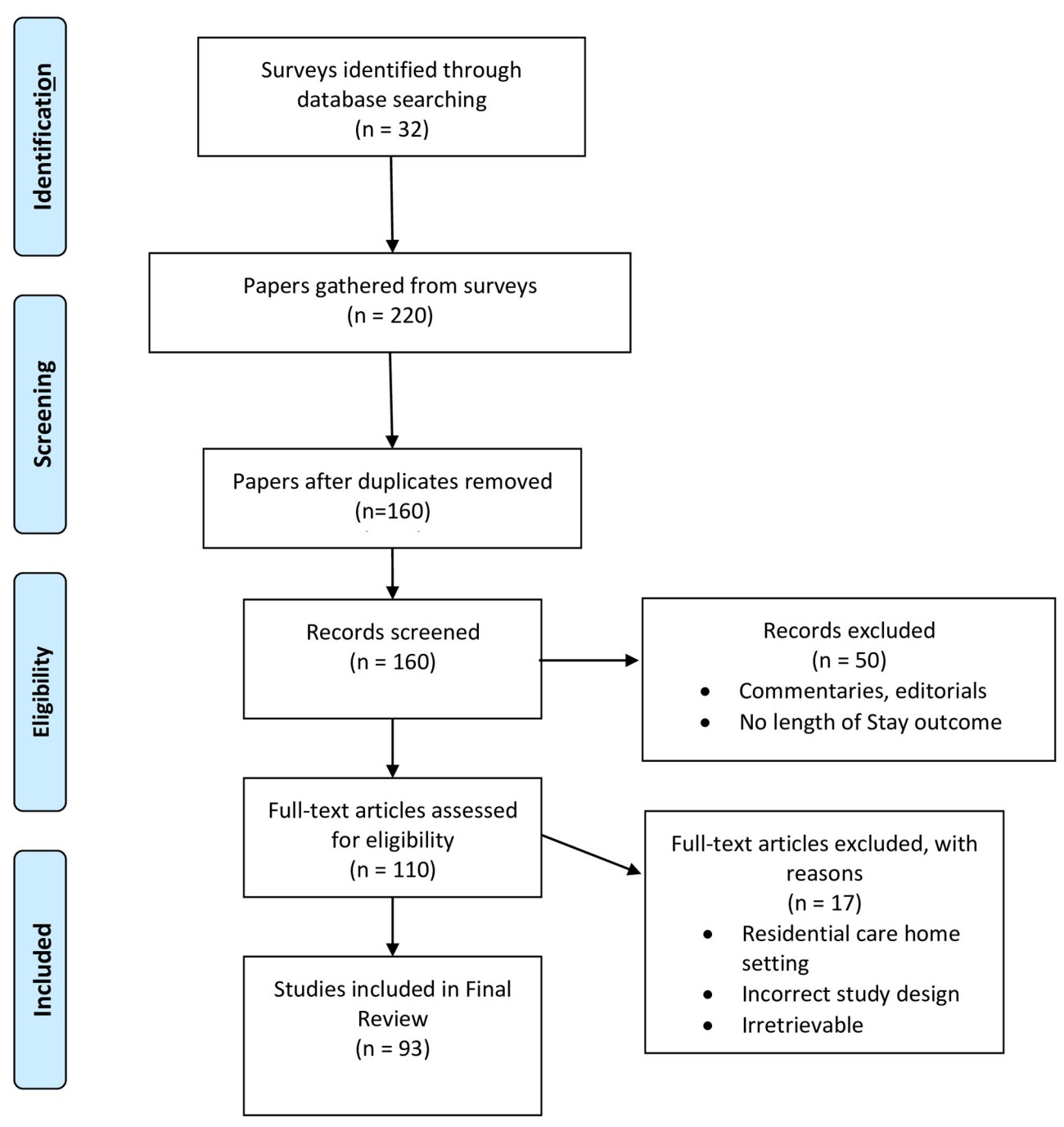

**Fig 1. PRISMA flow chart of the literature search, identification, and inclusion for the review.**

## 2.2 Study selection

The study selection for this systematic review was twofold: formulating appropriate eligibility criteria and abstract reading and selection for full-text reading. The following eligibility criteria was chosen for study inclusion:

1. Studies that deal with general adult LoS and specialised LoS.

2. Studies which examine one or more features that are related to patient LoS.

3. Studies that do not deal strictly with research in the medical. field related to clinical treatment.

4. Study that do not deal with experiments with animals.

5. Surveys and Systematic reviews of LoS prediction methods.

Any studies whose abstract or full-text did not meet any of the above criteria were excluded from this systematic review. This eligibility criteria was pilot tested randomly on 15 papers from the 93 studies that were chosen. Any disagreements were debated by at least three authors until an agreement about exclusion was reached. Studies that were not retrievable by electronic download were also excluded from this study as well as studies that were not written in the English language.

## 2.3 Data extraction

A data extraction sheet was developed and pilot tested on 15 randomly selected studies. The following data were recorded (when available):

1. Study characteristics (size of data cohort, year, continent, study design).

2. Patient characteristics (eligibility criteria).

3. Methods used.

4. Type of LoS measured (continuous or categorical).

5. Basic statistics presented (mean, standard deviation, significant/non-significant features, correlations, t-test, etc).

## 2.4 Quality assessment

Study quality was discussed using the Quality In Prognostic Studies (QUIPS) tool [12]. All parts of the QUIPS tool are considered including:

1. Study participation (studies that deal with general adult LoS and specialised LoS.

2. Study attrition (ensuring the cohort used is representative)

3. Feature measurement (the definition and measurement of features and the methods used for missing data should be robust)

4. Outcome measurement (the definition and measurement of LoS should be appropriate)

5. Methods, analysis and reporting (the model building should be appropriate and consider validation and does not selectively report results)

A rank was then included to assess a study based upon the above factors, this rank would vary from one to five based upon the number of factors that were satisfied.

## 2.5 Findings

As described in Fig 1, after removal of duplicate studies, 160 studies remained and were screened. All but 110 publications were excluded based on their content, article type and LoS outcome. The remaining 110 studies proceeded to quality assessment. At this stage 17 articles were excluded because it was deemed that they had poor study design, had a residential care home setting or were simply inaccessible. Therefore 93 studies were included in the final review. The characteristics of the included studies are summarised in Table 2. Each of these studies are described in the following sections of this review and are broke down by their respective approach type: (Operational Research-Based Approaches, Statistical and Arithmetic Approaches, and Machine learning and Data Mining approaches). The findings of this systematic review and survey are formulated as a discussion in section 6.

## 3 Approaches to modelling LoS

With the continued increase in healthcare costs, predicting the outcome of a serious illness or disease by estimating the LoS for a given patient is becoming increasingly important for the planning and assessment of interventions in a healthcare system [13]. These costs can be high due to a number of factors including the variety of drugs and therapies administered, the number of staff and the utilisation of large arrays of equipment in addition to the LoS of a patient. Swift identification of patients at higher risk of prolonged LoS or death will serve to significantly reduce these unavoidable costs, improve patient care and limit the likelihood of a patient succumbing to further healthcare complications [14]. A large proportion of LoS research has been predominantly focused on identifying the factors that strongly influence length of stay in different contexts as opposed to predicting the LoS outcome [15], [16]. There has been limited research that uses machine learning models which consider LoS directly. Current machine learning research tends to be concerned with specific patient cohorts and conditions [17], [18]. More recently, there has been a greater interest in novel deep learning approaches with healthcare environments placing increasing importance on the use of Electronic Health Records [19], [20]. This had led to the development of state-of-the-art predictive modelling approaches which aim to improve healthcare quality and increase personalised care across multiple clinical prediction tasks including LoS [21], [22].

It is also worth mentioning that there is considerable work in the literature which covers the problems of LoS prediction and mortality prediction. The prediction models for these problems use arithmetic models which make use of the mean and median as well as statistical modelling techniques such as regression analysis [23]. Data mining techniques have also been utilised in both areas. As mentioned previously, hospitals face continuing pressure to improve the quality of care provided to patients and reduce cost. This is common in intensive care units (ICUs) where the level of care required is considerably more complex, along with a higher associated cost. Hospitals assess the efficiency of care by measuring the hospital mortality rate and LoS in an ICU. It is for this reason that many of the models that have been proposed for the prediction of LoS are also applicable in the context of mortality prediction [24].

This section provides an overview and survey of the various analytical approaches that have been published in the LoS prediction literature in recent decades. A taxonomy of these different methods which is arranged according to the approach adopted, can be found in Fig 2, which are further discussed in subsequent sections.

### 3.1 Operational Research-Based approaches

One particular metric that is frequently used in the area of LoS prediction is the *average LoS*, which is relatively easy to calculate and quantify by the use of the mean. This mean is

**Table 2. A summary of the included LoS studies.**

| Paper Reference | Application Area | Approach Type |
|---|---|---|
| Huntley Et al. 1998 [2] | Psychiatry | Statistical and Arithmetic |
| Lim Et al. 2009 [3] | Inpatient Death | Statistical and Arithmetic |
| Chang Et al. 2002 [4] | Stroke | Statistical and Arithmetic |
| Yoon Et al. 2003 [15] | Emergency | Statistical and Arithmetic |
| Mezzich Et al. 1985 [16] | Psychiatric | Statistical and Arithmetic |
| Garg Et al. 2010 [64] | Stroke | Statistical and Arithmetic |
| Freitas Et al. 2012 [63] | Acute Care | Statistical and Arithmetic |
| Grubinger Et al. 2010 [65] | General LoS | Statistical and Arithmetic |
| Garg Et al. 2011 [62] | Stroke | Statistical and Arithmetic |
| Abelha Et al. 2007 [69] | Intensive Surgery | Statistical and Arithmetic |
| Caetano Et al. 2014 [70] | General LoS | Statistical and Arithmetic |
| Gruskay Et al. 2015 [71] | Spine Surgery | Statistical and Arithmetic |
| Newman Et al. 2018 [72] | Psychiatry | Statistical and Arithmetic |
| Toh Et al. 2017 [114] | Geriatric | Statistical and Arithmetic |
| Tu Et al. 1993 [75] | Cardiac | Intelligent Data Mining |
| Lowell Et al. 1994 [76] | Psychiatry | Intelligent Data Mining |
| Mobley Et al. 1995 [78] | Post-Coronary | Intelligent Data Mining |
| Rowan Et al. 2007 [79] | Cardiac | Intelligent Data Mining |
| Hachesu Et al. 2013 [80] | Cardiac | Intelligent Data Mining |
| Tsai Et al. 2016 [99] | Cardiology | Intelligent Data Mining |
| Azari Et al. 2012 [91] | General LoS | Intelligent Data Mining |
| Liu Et al. 2006 [94] | Geriatric | Intelligent Data Mining |
| Cai Et al. 2015 [97] | Mortality, Readmission and LoS | Intelligent Data Mining |
| Livieris Et al. 2018 [102] | General LoS | Intelligent Data Mining |
| Yang Et al. 2010 [17] | Burn Patients | Intelligent Data Mining |
| Stoean Et al. 2015 [18] | Colorectal Cancer | Intelligent Data Mining |
| Rajkomar Et al. 2018 [21] | General LoS | Intelligent Data Mining |
| Harutyunyan Et al. 2017 [22] | General LoS | Intelligent Data Mining |
| Shickel Et al. 2017 [19] | General LoS | Intelligent Data Mining |
| Yadav Et al. 2017 [20] | General LoS | Intelligent Data Mining |
| Stone Et al. 2019 [120] | Accident And Emergency | Intelligent Data Mining |
| Mulhestein Et al. 2018 [121] | Brain Tumour | Intelligent Data Mining |
| Cios Et al. 2002 [73] | General LoS | Intelligent Data Mining |
| Ahmad Et al. 2018 [83] | General LoS | Intelligent Data Mining |
| Holzinger Et al. 2017 [86] | General LoS | Intelligent Data Mining |
| Caruana Et al. 2015 [90] | Pneumonia | Intelligent Data Mining |
| Sundararajan Et al. 2004 [93] | Hospital Mortality | Intelligent Data Mining |
| Marshall Et al. 2001 [106] | Geriatric | Intelligent Data Mining |
| Cooper Et al. 1997 [124] | Pneumonia | Intelligent Data Mining |
| Cooper Et al. 2005 [125] | Pneumonia | Intelligent Data Mining |
| Mulhestein Et al. 2017 [128] | Brain Tumour | Intelligent Data Mining |
| Goldstein Et al. 2017 [130] | EHR | Intelligent Data Mining |
| Suresh Et al. 2017 [131] | Critical Care | Intelligent Data Mining |
| Avati Et al. 2018 [132] | Palliative care | Intelligent Data Mining |
| Awad Et al. 2017 [24] | General LoS | All |
| Millard Et al. 1994 [25] | Geriatric | Operational Research |
| Harper Et al. 2002 [26] | General LoS | Operational Research |

(*Continued*)

**Table 2.** (Continued)

| Paper Reference | Application Area | Approach Type |
| --- | --- | --- |
| Harper Et al. 2002 [27] | General LoS | Operational Research |
| Costa Et al. 2003 [28] | Critical Care | Operational Research |
| Millard Et al. 1988 [29] | Geriatric | Operational Research |
| Mcclean Et al. 1993 [30] | Geriatric | Operational Research |
| Mcclean Et al. 1993 [31] | Geriatric | Operational Research |
| Harrison Et al. 1991 [33] | Geriatric | Operational Research |
| Harrison Et al. 1994 [34] | General LoS | Operational Research |
| Mackay Et al. 2007 [35] | Acute Care | Operational Research |
| Millard Et al. 1996 [36] | Geriatric | Operational Research |
| Garcia-Navaro Et al. 2001 [37] | Geriatric | Operational Research |
| Millard Et al. 2001 [38] | Geriatric | Operational Research |
| Marshall Et al. 2005 [39] | General LoS | Operational Research |
| Brailsford Et al. 2001 [40] | General LoS | Operational Research |
| Cahill Et al. 1999 [41] | Critical Care | Operational Research |
| Davies Et al. 1995 [42] | General LoS | Operational Research |
| Gunal Et al. 2010 [43] | General LoS | Operational Research |
| El-Darzi Et al. 1998 [44] | General LoS | Operational Research |
| El-Darzi Et al. 2000 [45] | Aftercare | Operational Research |
| Vasilakis Et al. 2001 [46] | Wintercare | Operational Research |
| Davies Et al. 1994 [47] | General LoS | Operational Research |
| Irvine Et al. 1994 [48] | Geriatric | Operational Research |
| McClean Et al. 1978 [49] | General LoS | Operational Research |
| McClean Et al. 1998 [50] | Geriatric | Operational Research |
| Taylor Et al. 1997 [51] | Geriatric | Operational Research |
| Taylor Et al. 1996 [52] | Geriatric | Operational Research |
| Taylor Et al. 2000 [53] | Geriatric | Operational Research |
| Standfield Et al. 2014 [54] | General LoS | Operational Research |
| Guzman 2012 [56] | General LoS | Operational Research |
| Faddy Et al. 1999 [57] | General LoS | Operational Research |
| Xie Et al. 2005 [58] | Geriatric | Operational Research |
| Marshall Et al. 2002 [59] | Geriatric | Operational Research |
| Marshall Et al. 2003 [60] | General LoS | Operational Research |
| Robinson Et al. 1966 [112] | General LoS | Operational Research |
| Brameld Et al. 2006 [115] | General LoS | Operational Research |
| Chitnis Et al. 2013 [116] | Geriatric | Operational Research |
| Hall Et al. 2010 [118] | Rural Inpatient care | Operational Research |
| Scott Et al. 1993 [9] | Obstetric | Operational Research |
| Clarke Et al. 1996 [10] | General LoS | Operational Research |
| Bauer Et a. 2009 [13] | Discharge Planning | Operational Research |
| Shea Et al. 1995 [5] | General LoS | Operational Research |
| Butler Et al. 2018 [113] | General LoS | Operational Research |
| Russel-Weisz Et al. 2000 [117] | Rural Inpatient care | Operational Research |
| Wheatley Et al. 2007 [6] | Palliative care | Operational Research |
| Simmons Et al. 2005 [8] | Hospital overcrowding | Operational Research |
| Kenward Et al. 2004 [14] | Emergency Medicine | Operational Research |

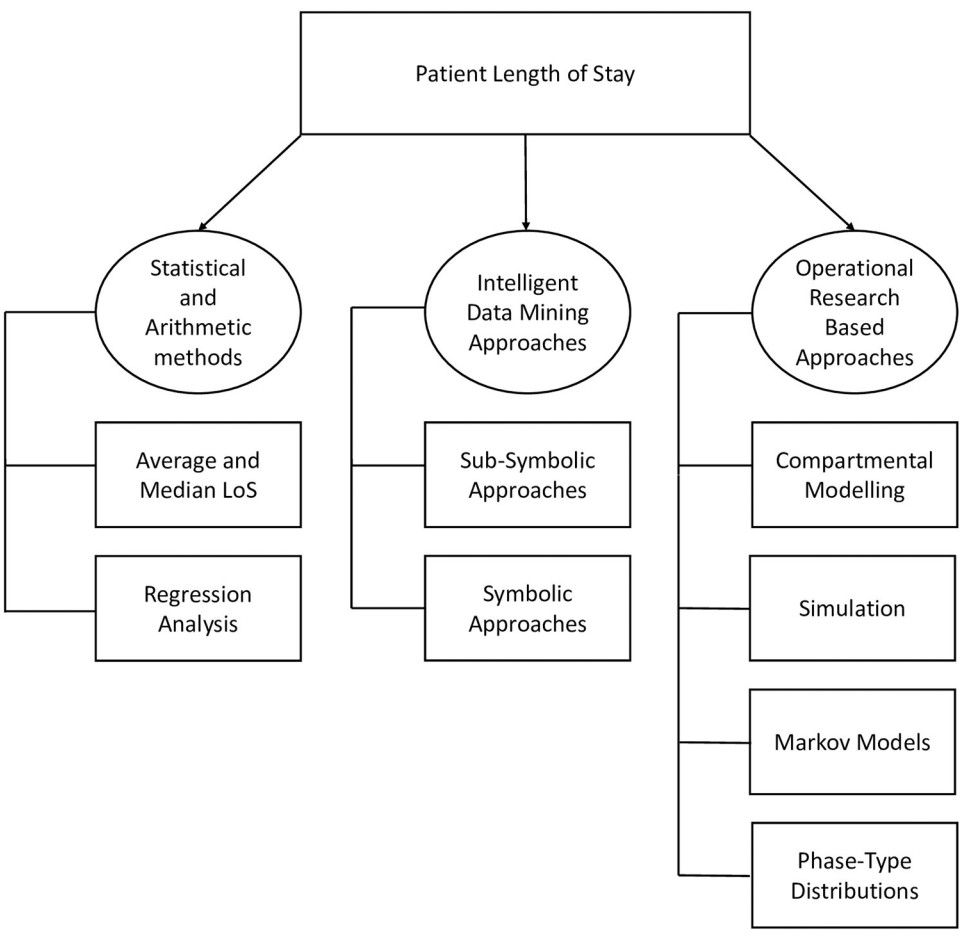

**Fig 2. A taxonomy of LoS prediction methods.**

calculated by dividing the sum of in-patient days by the number of patients admissions with the same diagnosis-related group classification. However, often it is not fully representative of the underlying data distribution, as the data can be very skewed [25]. It is commonly employed in more generic and simplistic models for basic planning and management of hospital capacity. Models that are derived using this metric tend to be very deterministic and take the form of spreadsheet based calculations [26]. Nevertheless, the hospital environment is (by its very nature) complex and uncertain, and as such straightforward simplistic approaches are not always useful [27]. The *average LoS* can be misleading if the underlying distribution of the data is non-normal and as a direct result, models that are built using *average LoS* could misinform by not accurately representing the patient population [28]. To address this issue, operational approaches have been developed which more effectively model patient flow and predict LoS.

In the following sections, four operational modelling based techniques employed to predict LoS are explored along with their applications.

**3.1.1 Compartmental modelling.** Compartmental systems are defined as being "systems which consist of a finite number of homogeneous, well mixed, lumped subsystems called compartments" [32]. Compartmental models can be linear, deterministic, non-linear or stochastic, depending on the process they have been designed to model. Over the past few decades

compartmental models have been applied to the way patients have been moved throughout various hospital systems.

In [29], it was observed that the LoS for geriatric departments could not be represented by a single metric such as the mean value. Instead the distribution of LoS matched a mixed exponential distribution. An exponential distribution is the probability distribution of the time between events and a mixed distribution is the probability distribution of a random variable derived from a collection of other random variables. In [33], this mixed exponential model is represented in the form:

$$N(s \geq x) = Ae^{-bx} + Ce^{-dx} \qquad (1)$$

where $s$ is the occupancy time of a patient, $x$ is the time in days, $N(s \geq x)$ is the total number of current patients who have stayed in hospital for greater than $x$ days. $A$, $b$, $C$ and $d$ are constants and parameters of the distribution.

In [33], the application of a compartmental flow model expanded the previously mentioned mixed exponential model. It is proposed that the movement of geriatric patients through the geriatric department can be modelled using two compartments, this is illustrated in Fig 3.

The model in Fig 3 displays patients who are initially admitted to an acute stay, from which they will die, be discharged at a rate of $R$ or be transferred to a long stay compartment a rate $V$. From the long stay compartment, the patient will either die or be discharged from the hospital at a rate of $D$. This model provides a method of estimating the number of acute and long stay patients and their estimated stay in hospital. It forecasts the average LoS as well as the average number of patients in each state. This two-compartment model aids healthcare professionals in effectively managing the use of beds in the geriatric department hence giving a valuable insight into both the patient flow around the department as well as the predicted LoS. The work in [34] later extended the two compartment model to incorporate a third compartment of *rehabilitative care* to represent patient flow in the hospital more accurately.

Compartmental modelling approaches focus on the use of a daily census where the parameters in use which describe the varying rates of flow are derived using the occupancy profile data from a single day [35]. One, two or three compartment statistics are generated depending on whether the best-fit mixed exponential equation has one, two or three components. This approach to modelling patient flow and LoS has been applied successfully in different areas of

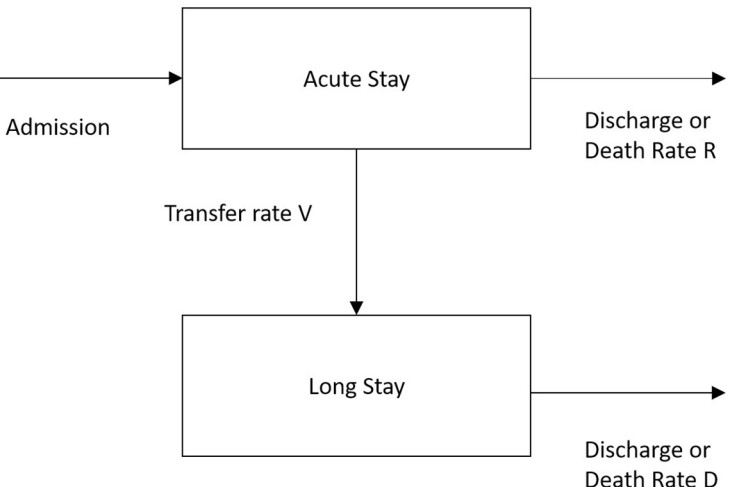

**Fig 3. Two-compartment model of a geriatric department.**

healthcare such as [36] and [37]. Notably, [38] analysed a one-night bed occupancy census containing records of 6,068 patients in seven different provider groups to model a health and social care system which included geriatric hospital beds, psychiatry beds and nursing homes.

Compartmental modelling is a well established and mathematically sound methodology that has been used consistently to model patient flow. The majority of compartmental modelling approaches have focused their efforts on specific patient cohorts such as geriatric patients, which is part of the reason that model uptake has been slow both by other LoS researchers and hospital decision makers alike. With the recent increased interest in "big data" and the expansion of efforts into investigating electronic health records it is likely that there will be a further decrease in the use of compartmental modelling in favour of more modern machine learning techniques such as artificial neural networks. The real disadvantage of compartmental models is that they are based on a single day census of beds and as such, they can be highly sensitive to the day the census was carried out. Consequently, it is unlikely that the LoS predictions will generalise well for an extended period. Also, compartmental models do not consider the cyclical effects of admissions and discharge and has no consideration of seasonality in the data.

**3.1.2 Simulation modelling.** One way of extending compartmental models is to view the problem as a queueing system. Queueing systems or simulations make use of performance measures such as "time in the system" and "time spent waiting in a queue", these measures can assist hospital planners in testing and preparing for particular scenarios and avoiding bottlenecks within the system. Discrete event simulation (DES) concerns the modelling of a system as it evolves over time by a representation in which the state variables change instantaneously at separate points in time, these are called events [39]. Events are occurrences that may change the state of the system. DES has been widely used in modelling healthcare systems [40]–[43].

Generally, the basic components of a patient simulation system are defined as follows:

- *Entities*: the elements of the simulated system such as patients.

- *Activities*: the operations and tasks that transform the entities such as compartments and queues.

- *The overall state*: a collection of features describing the system as a snapshot in time, such as the number of available beds, the number of patients in a queue, etc.

Such systems were first described in [44] where the development of the system via DES was used to perform numerical evaluations. The simulation model contained three compartments (*acute*, *rehabilitation* and *long-stay*). It was developed and tested against the results of established compartmental models. One of the findings from this work was the observation of a *"long warming up period"* during which the system would continually perform simulations until a steady-state was observed. This suggested that any change that was made to the system such as increasing beds or changes in patient LoS would require a prolonged period before the model would stabilise. Ultimately, by varying the policy parameters such as the overall level of emptiness, number of beds available, ward conversion rates and patient admissions, the simulation model could be used to assist hospital planners evaluate the effectiveness of a given geriatric department.

In general, the use of simulation modelling to support hospital managers allows not only for the potential to test changes to a system but also greater flexibility in gaining an understanding of the system that is being examined. This understanding is further expanded with the addition of external compartments such as independent home and support home to the basic model configuration [45]. The basic model was also configured to cater for a potential winter bed crisis scenario to attempt to determine the cause in English hospitals [46]. However, the application of these models in the real-world tends to be very operationally focused,

resulting in a need for large volumes of data to be captured. Furthermore, simulation models are only appropriate in the environment that they have been employed in. They are not generalisable, which means that whilst there is benefit to be gained from their use, the overall data requirements and development cost are likely to be an obstacle to general adoption by clinicians.

**3.1.3 Markov models.** Markov and semi-Markov models are models which make the assumption that sub groups of patients are homogeneous and as such, events occur at equally spaced intervals in time. These techniques can be useful in understanding aspects of patient flow and patient LoS, particularly in larger population groups where Markov assumptions can be applied [47]. A Markov model assumes a probabilistic behaviour of patients moving around a healthcare system and therefore provides a realistic representation of a real-world healthcare system.

The development of a continuous time stochastic model of patient flow is described in [48]. The paper details a two stage continuous time Markov model which illustrates the movement of geriatric patients through geriatric hospitals. The compartments in the model can be regarded as states and the probabilities of patients moving between those states can be computed (this could also be modelled using fuzzy inference systems—see alternative LoS prediction approaches). Patients that have been initially admitted to an acute stay state can be transferred to a long stay state or leave the hospital completely through either a death or a discharge state which is very similar to the aforementioned compartments for compartmental modelling. The work in [48] extends that of [33] which describes the distribution of LoS patients with a given census date by employing a mixed exponential distribution. The model in this case was deterministic and discrete time valued. This allowed for the estimation of the number of patients that were admitted to acute and long stay as well as their predicted LoS. As opposed to using discrete time, the work in [48] extends the approach to continuous time. By doing this, the approach in question can provide a means for calculating variances and co-variances for acute and long stay patients. Two different models are developed here; the first considers the situation in which there is a waiting list of patients. As such, the overall number of patients is assumed to be constant. The second model illustrates a situation in which admissions occur at random.

An extension to the previously mentioned stochastic Markov model was made to accommodate three stages [50]. It attaches a cost factor to each of the different stages which provides a model that can facilitate health and social services for the elderly whilst accounting for cost. Subsequently, in [51] the above approach is applied to a four compartment model in [52], where the four stages represent *acute*, *long-stay*, *community* or *deceased*. This model can estimate the expected number of patients at any time *t* in each of the stages for several patients which were admitted on the same day. Finally, in [53] these models were further extended to contain six stages with the premise of determining the underlying interactions between hospital medical services and community care.

Markov models are based on statistically sound methodologies and provide a useful approach to measuring and modelling patient flow. The models accurately reflect a given patient's journey through a healthcare system and give experts insight into potential hazards and the probabilities involved. Nevertheless, when compared with previously mentioned methods such as DES, it is less flexible, as DES allows for the modelling of interaction between patients and resources such as situations when constraints on resources mean that the choice of treatment for one patient affects what can be given to another. Markov models cannot accommodate these interactions [54]. Additionally, DES models can cater for higher levels of complexity in healthcare systems than Markov models, meaning that a generally greater level of detail and higher number of features can be captured. Markov models also require in-depth

knowledge of each of the Markov states of care in a hospital, this can be disadvantageous as more complex healthcare settings can sometimes contain a large number of different health states. This weakness can be alleviated through an expansion of the modelling process to incorporate conditional phase-type distributions which utilise prior knowledge of the process (described in more detail below). From a cost effectiveness perspective however, Markov models are a widely used, transparent method of modelling patient flow and LoS. Consequently, Markov models remain an important tool in assisting managers with resource management in hospital environments.

**3.1.4 Phase-type distributions.** Phase-type distributions were first introduced in [55]. They have been widely used in several different domains including queuing theory, drug kinetics and survival analysis. Phase-type distributions have the purpose of describing the time to absorption of a finite Markov chain in continuous time when there is only a single absorbing state and the stochastic process is transient [56]. These models are used to describe the duration until an event occurs, usually in the form of a sequence of phases or states of a Markov model. For example, in an LoS context, any transitions through transient states could correspond to the severity of illness that a patient is being treated for. This is because when a treatment ends (or reaches the absorption state) this could be associated with illnesses that are considered to be less severe as opposed to a much larger time until absorption which would constitute a more severe illness.

One of the major issues with using phase-type distributions to fit data is that they can be considerably overparameterised. This problem can mean that it is very difficult to give confidence intervals for various parameter estimates. In addition, these distributions tend to be very general, they can include distributions such as exponential, mixed exponential and continuous distributions. This generality can be an issue when attempting to make estimates for and analyse data as it can be difficult to identify and estimate the parameters of a phase-type distribution. As a remedy to this, a Coxian phase-type distribution was developed. A Coxian phase-type distribution is a unique type of Markov model which allows for the representation of the continuous duration of stay in hospital as a series of sequential phases which the patient will progress through until they leave the hospital [57]. An example of a Coxian phase-type distribution is shown in Fig 4. Here, phases $i = 1, 2...n$ are transient states of the process in question. Once a patient has transitioned through the necessary phases, they will reach an absorbing phase. $\lambda_1$ represents a transition between a phase to another phase and $\mu_1$ is the transition from a phase to the absorbing phase. A more formal definition of a Coxian phase-type distribution can be found in [57] and a practical example is explored in [58].

Conditional phase-type distributions are an expanded version of Coxian phase-type distributions which use Coxian phase-type distributions conditioned with a Bayesian belief network (BBN) of related features such as a patients characteristics that have an influence on a patient's given LoS. Conditional phase-type distributions are able to accommodate discrete and

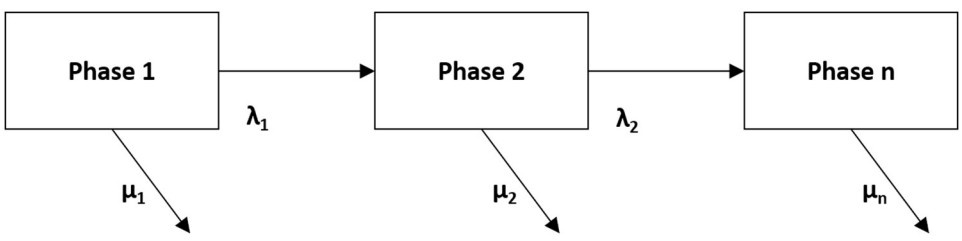

**Fig 4. Coxian phase-type distribution.**

continuous features and can represent highly skewed continuous distributions whilst also including causal information from relationships between features. Conditional phase-type distributions have been used to model the LoS of elderly patients and are explored in more detail in [59] [60].

Phase-type distributions may often provide detailed and accurate results for LoS analysis and their use is accepted in the statistics community. The phases that are employed by this type of distribution are easy to interpret by non-mathematicians because LoS data often consists of phases that reflect the properties of hospital environments in the real world such as long-stay or short-stay wards [39]. The main problem with such distributions is that their implementation tends to be ad-hoc in nature, the vast majority of work is only performed by researchers in this field. Additionally, while these approaches can produce valuable insights into the causality of patient LoS and patient cost, the construction methodology is much more complex than more traditionally employed methods. This means that further work is required to ensure that the model output is clinically meaningful. Furthermore, phase-type distributions have almost exclusively been used in data consisting of elderly patient populations, consequently it cannot be assumed that they would perform well when applied to other populations or contexts.

## 3.2 Statistical and arithmetic approaches

Statistical methods such as logistic regression which include the analysis of features are used in the prediction of LoS. In the context of LoS, the features would be defined as a patient's characteristics and the associated external factors which could influence the prediction of LoS. Models that utilise regression analysis typically make use of a patient's diagnosis/diagnoses, procedures, gender and age [62]. Regression modelling has been used to examine the relationship between various administrative features such as *year of discharge*, *comorbidities* and *age*, from in-patients receiving care at an acute care hospital in Portugal [63]. It was discovered that *age*, *type of admission*, and *hospital type* were significantly associated with high LoS outliers. It was also found that the high proportion of LoS outliers in the dataset does not seem to relate to a patient's financial coverage.

To provide a more complete analysis, it is often necessary to manually inform a generated regression tree model with medical knowledge. However, it is possible that this can reduce the overall predictive accuracy of successive models [64]. In [65] the case is made that any small change to the data can result in an entirely different tree, even though each tree will be statistically accurate. Consequently, [65] utilised bootstrap-based *model bumping* [66]. Generally, bootstrap methods are used in an attempt to reduce the amount of prediction error by averaging and combining various models, examples of bootstrap methods would be boosting [67] and Random Forests [68]. This bootstrap-based method was used to build a series of more diverse regression tree models through systematic re-sampling of the data. The conclusion of this work was that model bumping was a very powerful tool for constructing accurate regression trees that could be used as models for diagnosis related group (DRG) systems.

In 2007, an evaluation of LoS of patients submitted to a non cardiac surgery and admitted to a surgical Intensive Care Unit between 2004 and 2005 was performed [69]. The features that were used to categorise the patients were: *age, gender, body mass index, physical status, type, magnitude of surgery, duration of anesthesia, temperature on admission and LoS in the hospital*. The work made use of a linear regression model which showed that the average LoS was 4.22 days.

Another instance in which regression analysis was adopted for the task of LoS was on a case study of a Portuguese hospital which used a large dataset collected between 2001 and 2013

[70]. A regression approach is employed, and several state-of-the-art data mining models were compared in order to predict the real number of LoS days as opposed to a typical classification task. The best results were achieved by using by a Random Forest regression which revealed that the physical service where the patient was hospitalised, and the associated medical speciality were more important in determining LoS than the patient's attributes such as *age*, *education* and *sex*. The explanatory and predictive power that is yielded by such an approach is very useful for decision-support for medical professionals at healthcare institutes.

In 2015, a study was carried out to investigate the factors that influence LoS after elective posterior lumbar spine surgery [71]. 103 patients who underwent elective, open or one to three level posterior lumbar instrumented fusion by the orthopaedic spine service between 2010 and 2012 were included in the study. The study investigated preoperative factors such as *patient demographics*, *previous surgery* and *comorbidities* as well as intraoperative factors such as *estimated blood loss* and *fluids administered* in addition to postoperative factors such as *blood transfusion complications*. These factors were then analysed via the use of multivariate stepwise regression to determine the main predictors of LoS. The study found that the average LoS was 3.6 ± 1.8 days. Patients who were older and had widespread systemic disease tended to stay in the hospital for prolonged periods after surgery with a *history of heart disease* being associated with a shorter LoS.

More recently in 2018, a study was performed to identify the factors that were associated with prolonged stays in a psychiatric hospital in a UK mental health trust [72]. Multiple regression and a multiple imputation method (to deal with missing data) were used to analyse 7,653 in-patient admissions. The factors that were discovered to have been associated with a longer LoS were *gender, ethnicity, accommodation status* and *a primary diagnosis of psychosis* as well as *the number of care coordinators* that were present at the hospital.

Regression approaches are amongst the most widely used in the healthcare domain particularly in the area of LoS prediction. There is still work being published to date which addresses LoS prediction for a wide range of different patient cohorts and conditions. These approaches are used to model complex non-linear relationships between different independent and dependent features which have an impact on a patient's LoS. Regression techniques are already disseminated to a much wider audience, and as such have the trust and acceptance of medical clinicians. In the context of LoS, regression appears to remain the most popular choice as it allows the modeler to easily understand the effect of features on the outcome LoS. However, regression models often require their developers to undertake formal statistical training and have an understanding of a variety of complex statistical concepts and therefore more time may need to be invested in order to utilise regression-based methods in a effective practical manner.

### 3.3 Machine learning and data mining approaches

In addition to regression analysis, data mining approaches to LoS prediction have also become more popular. A data mining approach refers to an approach that is designed to extract usable data from a larger set of raw data. These approaches make use of techniques such as clustering and classification to perform knowledge discovery from data. Although there is some overlap with the studies mentioned previously, particularly [70], these techniques aid the user in discovering patterns in large datasets by uncovering the hidden patterns of various relationships in the data. It is from these relationships that knowledge can be extracted that can support decision making in a hospital or clinical environment. This is more commonly known as medical data mining [73]. As previously mentioned, there has been limited machine learning research that directly considers the goal of predicting LoS, instead the currently adopted

machine learning methodologies focus on patients with specific conditions or the work centres around the factors that influence LoS in different contexts. In this section, the symbolic and sub-symbolic approaches [74] that have been employed for the task of LoS prediction are explored.

**3.3.1 Sub-symbolic approaches.** Sub-symbolic learners such as Neural Networks (NNs) and Support Vector Machines (SVMs) have been used in a wide range of applications. They have the ability to implicitly identify relationships between a series of independent features and their corresponding dependent features that would have been otherwise unattainable. In the last 30 years, there has been an increased interest in sub-symbolic approaches in a clinical setting particularly in the area of predicting LoS.

In 1993, an approach utilised a NN for predicting ICU length of stay following cardiac surgery [75]. The population consisted of 1,409 patients who underwent open heart surgery in Toronto. The network was able to effectively divide patients into three heterogenous groups: *low, intermediate and high risk* of prolonged LoS. The overall performance of the network was evaluated using the area under the Receiver Operating Characteristic curve (ROC) and was found to be 0.7094 in the training set and 0.6960 in the test set. It was concluded that the network was able to perform to a satisfactory level but required further prospective clinical testing to determine whether or not it would be clinically useful.

In 1994, similar work which tested the effect of diagnosis on training a NN to predict the LoS for psychiatric patients involuntarily admitted to a state hospital was carried out [76]. A series of NNs were trained which represented Schizophrenia, affective disorders and diagnosis-related group. The features that were used to train the networks included a patient's demographics, severity of illness, and nature of residence amongst other features that were identified to be significant in assessing a patient's LoS. The NN predictions were compared with actual LoS indicated accuracy rates which ranged from 35% to 70%. The validity of these predictions were measured by comparing the LoS estimates with a clinical treatment team's predictions at 72 hours after admission. In all cases, the NN was able to predict as well as or better than the treatment team.

Equally, in [78] the LoS of patients receiving care at a post-coronary care unit to predict possible stays of 1 to 20 days was investigated. An ANN [77] was trained using 629 patient records and used 127 records as a test set. There was an average 1.4 day difference per record between the actual LoS in the test set and the predictions of the network. The actual LoS predicted within 1 day was predicted with an overall accuracy of 72%. It was concluded that ANN-based classifiers demonstrated an ability to utilise common patient admission characteristics as predictors for LoS.

In a further study [79], ANN-based learners are also employed in order to stratify the length of stay of cardiac patients into risk groups based upon preoperative and initial postoperative patient feature values. The work focused on 1,292 patients that underwent cardiac surgery between 2001 and 2003 in the department of cardiothoracic surgery. Reviewing contemporary literature, 15 preoperative risk factors and 17 operative and postoperative features were discovered to have had an influence on LoS. ANNs and ensembles of ANNs were applied to the scaled data. The study concluded that ensembles of ANN-based learners were best suited to the task of predicting LoS for postoperative cardiac patients compared with ANN-based learners in isolation.

In [80] three different learners were applied to the task of classifying the LoS of coronary patients. These three techniques were drawn from different areas of machine learning, namely: Decision Trees [81], Support Vector Machines (SVMs) [82] and ANNs. The data consisted of the patient records of 4,948 patients who had suffered from coronary artery disease. The data included 36 different features. The dataset was partitioned into a training set and a testing set:

80% of the data was used for training and 20% of the data was used for testing. The training set was used to select the optimal hyperparameters of the models and the testing set was used to evaluate each model's predictive ability. This study determined that all three algorithms were able to predict LoS with varying degrees of accuracy with SVM scoring the highest at 96.4%. It was also revealed that there was a strong tendency for LoS to be longer in patients with lung or respiratory disorders and high blood pressure. It is important to note however that despite the SVM algorithm outperforming the others, it can be very difficult to understand the underlying rules that are learned by sub-symbolic techniques as opposed to symbolic based learners such as decision trees or logistic regression where the output is transparent to human scrutiny. This aspect is often of paramount importance for medical applications and diagnosis as human expert input is often used to assist in understanding the data and the underlying relationships between the features and the decision classes [83].

Two stage LoS prediction was utilised for predischarge and preadmission patients in [99]. The predischarge stage makes use of all of the available data for hospital in-patients and the preadmission stage uses only the data that is available prior to a patients admission. The overall prediction results of the predischarge patients were used to evaluate the LoS prediction performance at the preadmission stage. The data set contained records of 2,377 cardiovascular disease patients with one of three diagnoses: Heart failure (HF), Acute Myocardial Infarction (AMI) and Coronary Atherosclerosis (CAS). The generated classification model was able to correctly predict 88.07% to 89.64% with a mean absolute error (MAE) of $1.06 \sim 1.11$ at the predischarge stage and 88.31% to 89.65% with an MAE of $1.03 \sim 1.07$ at the preadmission stage, respectively for CAS patients using an ANN. For HF and AMI patients the prediction accuracy ranged from 64.12% to 66.07% at the predischarge stage with an MAE of $3.83 \sim 3.91$ and 63.69% and 65.72% with an MAE of $3.87 \sim 3.97$ at preadmission.

In 2020, a study provided an accurate patient specific risk prediction for one-year postoperative mortality or cardiac transplantation and prolonged hospital LoS with the purpose of assisting clinicians and patient' families in the preoperative decision making process [98]. The study applied 5 Machine learning algorithms (Ridge logistic regression, decision tree, random forest, gradient boosting, deep neural network) which predicted and calculated individual patient risk for mortality and prolonged LoS using the Pediatric Heart Network Single Ventricle Reconstruction Trial dataset. A Markov Chain Monte-Carlo simulation method was used to impute missing data and the feed the features to the machine learning models. The deep neural network model demonstrated 89 ± 4% accuracy and 0.95 ± 0.02 AUCROC.

Similarly, in 2020, a study proposed a 2 general-purpose multi-modal network architectures to enhance patient representation learning by combining sequential and unstructured clinical notes with structured data [100]. The proposed fusion models leverage document embeddings for the representation of long clinical note documents and either convolutional neural network or long short-term memory networks to model the sequential clinical notes and temporal signals, and one-hot encoding for static information representation. The performance of the proposed models on 3 risk prediction tasks (hospital mortality, 30-day readmission and long LoS prediction) was evaluared using derived data from the MIMIC III dataset. The results showed that by combining unstructured clinical notes with structured data, the proposed models outperform other models that utilize either unstructured notes or structured data only.

As recently as 2021, a study assessed the effectiveness of machine learning models using daily ward round notes to predict the likelihood of discharge within 2 days and predict the likelihood of discharge within 7 days as well produce an estimated date of discharge on a daily basis [101]. Daily ward round notes and relevant discrete features were collected from the electronic medical record of patients admitted under General Medicine at the Royal Adelaide hospital over an 8-month period. Artificial neural networks and logistic regression were effective

at predicting discharge within 48 hours of a given ward round note. These models achieved AUC of 0.80 and 0.78, respectively. Prediction of discharge within 7 days of a given note was less accurate, with artificial neural network returning an AUC of 0.68 and logistic regression an AUC of 0.61.

The inherent success of sub-symbolic learners is irrefutable; they perform well in identifying complex, non-linear relationships in the data. Nevertheless, the "black-box" nature of these techniques is an obstacle to acceptance by clinicians and medical experts in a hospital environment. It is likely that clinicians will be reluctant to welcome the achievements of these approaches despite the benefits their predictive abilities might bring, as there is no explicit explanation for the derivation of their results. Inevitably, this calls for systems that support decisions which are explainable and transparent, especially with the rise of legal and privacy legislation in the form of the European General Data Protection Regulation (GDPR) which could make justifying the use of blackbox approaches more difficult. As such, research towards building *explainable-AI (XAI)* systems has become increasingly prevalent, particularly in a medical domain [86]. Explainable models typically take two forms, Post-hoc systems and Ante-hoc systems. Post-hoc systems provide local explanations for a specific decision and make it reproducible on demand. An example of this is BETA (Black Box Explanations through Transparent Approximations) which is a model-agnostic framework for explaining the behaviour of a given black-box classifier by optimising for high agreement between the original model and general interpretability of the explanation of the model, first outlined in [87]. Ante-hoc systems however, are interpretable by design and have been termed "glass box approaches" [88]. Examples of such approaches include decision trees and fuzzy inference systems. Fuzzy inference systems in particular, have historically been designed from expert knowledge or data and demonstrate a valuable framework for interaction between human expert knowledge and hidden knowledge in the data [89]. A further example of XAI is the use of high performance generalised additive models with pairwise interactions (GAMs) which were applied to the medical domain and can yield explainable and scalable models with a high degree of predictive accuracy on large datasets [90]. Currently in the domain of LoS prediction, there is a lack of work which makes use of explainable sub-symbolic learners, however, the increasing widespread applicability of these models necessitates a need for explanations in order to hold sub-symbolic approaches accountable in healthcare environments [83].

Furthermore, modelling length of stay using deep learning can lead to some ethical implications. Even a healthcare task as simple as determining whether a patient has a disease can be skewed by how prevalent diseases are, or how they are manifested in specific patient populations. One example of this would be a model that has been created to predict if a patient will develop heart failure. This model will undoubtedly require patients who have heart failure and those patients without heart failure. The selections of these patients can often rely on parts of EHR data that can be skewed due to either the lack of access to care or abnormalities in clinical care. More specifically, clinical protocol can affect the frequency and observation of abnormal tests [84] and naive data collection can yield inconsistent labels in chest X-rays [85]. Biased labelling and the models that result from this labelling can be a crucial factor when it comes to clinical resource management of a healthcare system.

In the same way, developers of clinical LoS models may also choose to predict healthcare costs. This means that a machine learning model will seek to predict which patients will have a prolonged LoS in hospitals and which patients will cost the healthcare provided more in the future. Some model developers may use healthcare costs as a proxy for future health needs to guide accurate healthcare interventions for high cost patients. However, it also possible that other modellers may explicitly want to understand patients who will have a high healthcare cost to reduce the total cost of healthcare. This could lead to healthcare providers denying care

to those perceived to have a prolonged LoS in hospital. This could potentially be a worrying trend and because socioeconomic factors affect both access to financial resources and health-care, these models could yield predictions that exacerbate inequities.

As well as a considering the ethical implications of using machine learning models for EHR Data, it is also important to consider the opportunities for Bias in using EHR data for machine learning in clinical decision support. As the utilisation of machine learning in healthcare is increasing exponentially, the underlying data sources and methods of data collection should undergo examination. As mentioned previously, it is possible that these modelling approaches could worse or perpetuate existing health inequalities. There is no doubt that any observational study or statistical modelling method could succumb to bias; however, the data that is available in healthcare has the potential to affect important clinical decision support tools that are based on machine learning. One example of such bias is missing data bias. EHR data may only con-tain more severe cases for specific patient populations and make incorrect inferences about the risk for such cases. Incomplete data can result in large portions of the population being elimi-nated and result in inaccurate predictions for certain patient groups. To be clear, this could affect vulnerable patient populations and as such could lead to patients having more fractured care and or being seen at multiple or varying healthcare institutions.

Measurement error is another important bias that could be apparent in EHR data. For example, patients of a lower socioeconomic status may be more likely to be seen in teaching clinics as opposed to private care where the data input or clinical reasoning could be less accu-rate or different from that of patients from a higher socioeconomic status. This implicit bias could lead to disparities in the level of care provided to patients with different socioeconomic backgrounds. A machine learning model that contends with healthcare data that is collected in this environment could inaccurately learn to treat patients of low socioeconomic status according to less than optimal care or according to the implicit bias of the data.

**3.3.2 Symbolic approaches.** In [91] an approach for the prediction of hospital LoS using multi-tiered data mining is presented. Multi-tiered data mining is described to be a methodol-ogy that employs $\mathbb{K}$-means clustering [92] to generate the training sets to train 10 different classification algorithms. The performance of the different algorithms is compared using a wide range of performance metrics. $\mathbb{K}$-means clustering was applied to the raw data and the selected $\mathbb{K}$ value was based on three different criteria. The number of coded medical condi-tions that the data contained (44), the number of Charlson index codes [93] in the data (a cate-gorised comorbidity score, 4) as well as determining $\mathbb{K}$ by trying to minimise it's Sum of Squared Errors (SSE), which yielded a $\mathbb{K}$ of 9. The data partition for the cluster with SSE at a minimal value resulted in an increase in overall performance with the J48 algorithm having an accuracy of 72.368% and an AUC of 0.813. The work provided valuable insights into the underlying factors that influence hospital LoS as it was able to identify patients that would require aggressive or moderate early intervention to avoid longer stays in hospital.

In [94] several decision tree algorithms and Naive Bayes classifiers were applied to a geriat-ric hospital dataset, (commonly referred to as the *Clinics Dataset* or the *Millard Dataset* in the literature). A Naive Bayes classifier is a classification technique based on Bayes' Theorem [95] which assumes independence among features—it assumes that the presence of specific feature in a class is unrelated to the presence of any other feature. The dataset contains 4,722 patient records containing feature information related to admission and discharge dates, reason for admission and the overall outcome. This study attempted to predict the in-patient LoS for pro-longed stay patients. To handle the considerable amount of missing data within the dataset the paper makes use of Naive Bayes Imputation (NBI) [96]. A decision tree is then constructed on both the imputed and non-imputed data and the classification accuracies are compared. The

NBI models considerably increased classification performance of predicting LoS, but this data is of course manipulated and therefore may be biased.

Further work was performed to predict the LoS of patients in real time. A Bayesian network model was built to estimate the probability of a hospitalised patient being "at home", "in the hospital" or "deceased" for the following 7 days [97]. Bayesian networks are a probabilistic graphical model that capture the known conditional independence between features in the form of directed edges in a graph model [61]. Electronic health records of 32,634 patients admitted to a Sydney metropolitan hospital via the emergency department from 2008 to 2011 were used. The model achieved an average daily accuracy of 80% and an Az of 0.82. It was able to predict at the highest rate within 24 hours from prediction and decreased slowly over time. The model at the time was the first non-disease specific model that concurrently predicted remaining days of hospitalisation, death and readmission in the same outcome. The conclusion is that Bayesian networks can effectively model electronic health records to provide real-time forecasts for patient pathways which support better decision-making in healthcare systems.

In 2018, [102], semi-supervised learning was employed in place of supervised learning as it was argued that they have become more prominent and tend to exhibit good performance over fully labelled data but lack the ability to be applied to a large amount of unlabelled data. The performance of semi-supervised algorithms are assessed in predicting the length of stay of hospitalised patients. The work compared three different semi-supervised learning algorithms Self-training, Co-training, Tri-training using 10-fold cross-validation on four different proportions of the labelled training data of 10%, 20%, 30%, 40%. Tri-training was found to be the most effective method as it achieved the best results with the 20% and 40% proportions across all five base learners, achieving a superior performance in five cases for the 20% proportion and better performance in four cases for the 40% proportion of labelled data. These results were then statistically verified using a Friedman aligned ranking statistical test. The results indicated that semi-supervised learning can improve the classification accuracy using fewer labeled and more unlabeled data for developing reliable LoS prediction models.

In 2020, a study investigated factors contributing to long-term hospitalisation of schizophrenic offenders referred to a Swiss forensic institution, using machine learning algorithms to detect nonlinear dependencies between features [103]. It was a retrospective study that included the notes of 143 Schizophrenic offenders categorised by their 90 features. In order to quantify the influence of the 90 features on the model and to reduce the algorithm's susceptibility to overfitting forward selection was used [104]. Forward selection is a technique which is based on subset selection and is a statistical regression method to find a small subset of available feature that are the most relevant to the LoS outcome. This resulted in nine features that were then ranked according to their importance alongside statistical significance tests. Two factors were identified as being particularly influential for a prolonged forensic hospital stay, both of which were related to the aspects of the index offence, (attempted homicide, extent of the victim's injury).

Finally, a novel simple mortality risk level estimation system that can determine the mortality rate of a patient by combining LoS and age was developed [105]. In this study, a combination of just-in-time learning (JITL) and a one-class extreme learning machine (ELM) is proposed to predict the number of days a patient stays in hospital. Where JITL is used to search for personalized cases for a new patient and one-class ELM is used to determine whether the patient can be discharged within 10 days. This resulted in the model having AUC of 0.8519, a lift value of 2.1390, an accuracy of 0.82 and specificity and sensitivity of 1 and 0.6150 respectively.

## 4 Data used for LoS

LoS datasets vary widely in terms of the number of records they contain and the number of features they consider. This is often due to a variety of reasons including the ease and cost of data collection, as well as the patient cohort that data collection is focused upon. The size of these datasets depends on the availability of electronic records within the hospital in question. Electronic records allow for easy manipulation and cleaning of the data as well as a large number of total LoS patient records. Generally, the larger LoS datasets are drawn from dedicated healthcare service databases that have been collecting data for prolonged periods of time. On the other hand, studies in the literature that perform manual data collection have fewer patient records or fewer features to select from at the initial data preparation stage. Moreover, a manual data collection can be susceptible to human error resulting in an inherent noise level in the data. The variation in size and purpose of these datasets can make it difficult to compare the relative success of each of the approaches that are applied to LoS datasets. This has led to ad-hoc methods being employed as the data that is being collected is only relevant to the hospital or health authority it was collected in. This means that any generated prediction models can also only be employed in that particular hospital. It is because of this that there is no standard methodology for predicting LoS and many of the decisions that are made to alter the data are made using expert domain knowledge or in an effort to enhance the accuracy of prediction or in some cases are performed without justification.

In this section, a series of different datasets that have been used in the literature are examined. The features that were chosen in each dataset are discussed as well as the data preprocessing procedures that were employed for the datasets. The Clinics dataset is the first dataset to be considered, It contains data from a clinical computer system that was in use between 1994 and 1997, for the management of patients in a Geriatric Medicine department of a metropolitan teaching hospital in the UK [106]. The Clinics dataset is the most widely used dataset in the literature for LoS prediction. The remaining two datasets, have only been used in their relevant studies, the first of which is a study using the data from a Portuguese hospital in Lisbon [70]. The data was collected from the hospitalisation process between 2000 and 2013 which included 26,462 records from 15,253 patients. The data was used with the purpose of predicting generic LoS for all hospital services using a pure regression approach which predicts the actual number of LoS. The final dataset that will be examined will be a dataset from a hospital in Kalamata, Greece [102]. The data was collected from 2008 to 2012 and identified 4,403 patients over the age of 65 years. The premise of the study was to evaluate the performance of semi-supervised learning methods in predicting the length of stay of hospitalised patients. All of the datasets that were chosen with the exception of the Clinics dataset have all utilised machine learning and AI based approaches in order to predict LoS. A summary of the datasets is provided in Table 3.

### 4.1 Clinics dataset

The Clinics dataset contains 4,722 records which included patient attributes, admission reasons, discharge details as well as the outcome and duration of stay, a summary of the dataset is

**Table 3. Summary of datasets.**

| Dataset Name | No. of Features | No. of data objects | Decision feature |
|---|---|---|---|
| *Clinics* | 15 | 4722 | LoS and discharge destination |
| *Lisbon Portugal* | 14 | 26462–15253* | LoS |
| *Kalamata Greece* | 12 | 4403 | LoS |

*26,452 Patient in-patient episodes with 15253 patients

**Table 4. A summary of the Features of the clinics dataset.**

| Clinics Dataset | | |
|---|---|---|
| **Feature** | **Feature type** | **Values** |
| *Age* | discretised | {< *85 years,* ≥ *85 years*} |
| *Gender* | binary | *Male/Female* |
| *Admission Year* | integer | *(1994, 1995, 1996, 1997)* |
| *Lives Alone* | binary | *Yes/No* |
| *Marital Status* | categorical | *Widowed/Single/Married/Other* |
| *Admission Method* | categorical | *Emergency Admission/Emergency GP/Transfer/Planned Admission* |
| *Season* | categorical | *Spring/Summer/Autumn/Winter* |
| *Stroke* | binary | *Yes/No* |
| *Fall* | binary | *Yes/No* |
| *Decreased Mobility* | binary | *Yes/No* |
| *Confusion* | binary | *Yes/No* |
| *Urine Incontinence* | binary | *Yes/No* |
| *Secondary Reasons* | binary | *Yes/No* |
| *Barthel Score* | intervalised | {*1, 2–10, 11–19, 20+*} |
| *Destination* | categorical | *deceased/home/transfer* |
| *Duration of Stay* | categorical | {*0–17, 18–49, 50+*} |

shown in Table 4. Preparation of the dataset before the analysis required the categorisation and regrouping of some of the features such as *age*, *marital status* and *the duration of stay*. As such, the analysis of relationships between the features and a patient's outcome was enabled. The Barthel score is composed of various indices that assess a patient's ability to perform every day activities and their dependence on others for support, it consists of ten elements: *feeding, grooming, bathing, mobility, stairs, dressing, transfer, toilet, bladder* and *bowels*. Each patient is assessed using a scale of dependency ranging from 0 to 3. A low score would constitute a high level of patient dependency while a high score would reflect a patient who is independent. For this dataset, the scoring system was simplified and a grouped Barthel score is used: *heavily dependent* (Barthel score: ≤1), *very dependent* (Barthel score: 2 − 10), *slightly dependent* (Barthel score: 11 − 19) and *independent* (Barthel score: ≥20). The length of stay decision feature was discretised into *0–17 days*, *18–49 days* and *50+ days*. This grouping was arrived at in agreement with clinical input. Each group was intended to model the stages of survival in a hospital, *acute stay*, *rehabilitation* and *long stay*. The clinics dataset also has a large proportion of missing values, 3,017 patient records out of 4,722 (63.89%) contain missing values. Overall patient LoS in the dataset has an average of 85 days and a median of 17 days.

## 4.2 Lisbon Portugal dataset

A summary of this dataset is shown in Table 5. Initially, the statistical package R [109] was adopted for the purpose of exploratory analysis of the data, this included the use of histograms and boxplots. During this stage, outliers were detected; One record with an LoS of 2,294, an age of 207 and 20 other records related with a virtual medical speciality were removed from the data. Following this step the data contained 26,431 records. Data preprocessing was then performed on the data which involved feature selection, handling data with missing values and appropriate feature transformations. Fourteen features from the above table were discarded during the selection of features. *Date of Birth* was discarded as it was reflected in the age feature. *Country* was removed because 99% of the patients were of Portuguese nationality.

**Table 5. A summary of the Lisbon Portugal dataset.**

| Lisbon Portugal Dataset | | |
|---|---|---|
| **Feature** | **Feature type** | **Values (Description)** |
| *Sex* | binary | *Male/Female* |
| *Date of Birth* | date | *Value* |
| *Age* | intervalised | $\{\leq 15, 15–44, 45–64, 65–84, \geq 85\}$ |
| *Country* | nominal | *Residence country* |
| *Residence* | nominal | *Place of residence* |
| *Education* | categorical | Educational attainment |
| *Marital Status* | categorical | *Marital status* |
| *Initial Diagnosis* | ordinal | Initial diagnosis description |
| *Episode Type* | nominal | *Patient type of episode* |
| *Inpatient Service* | nominal | *Physical inpatient Service* |
| *Medical Speciality* | nominal | *Patient medical Speciality* |
| *Admission Request Date* | date | *Date for hospitalisation admission request* |
| *Admission Date* | date | *Hospital admission date* |
| *Admission Year* | ordinal | *Hospital admission year* |
| *Admission Month* | ordinal | *Hospital admission month* |
| *Admission Day* | ordinal | *Hospital admission day of the week* |
| *Admission Hour* | ordinal | *Hospital admission hour* |
| *Main Procedure* | categorical | *Main procedure description* |
| *Main Diagnosis* | categorical | *Main diagnosis description* |
| *Physician ID* | nominal | *Identification of the physician responsible for the internment* |
| *Discharge Destination* | nominal | *Patient destination after hospital discharge* |
| *GDH* | categorical | *Homogeneous group diagnosis code* |
| *Treatment* | categorical | *Clinic codification for procedures, treatments and diseases* |
| *GCD* | categorical | *Great diagnostic category* |
| *Previous Admissions* | numeric | *Number of previous patient admissions* |

*Residence* was removed because a large percentage of this feature was missing in addition to the sheer volume of nominal levels that existed within the feature domain. Half of the *Admission Request Date* feature consisted of missing values and this information was mostly reflected in the *Admission Date* and as such was removed. *Admission Date* was discarded as it was featured in the *Admission Month*, *Day* and *Hour* as well as LoS. *Admission Year* was discarded as it was deemed not relevant. *Physician ID* had a large percentage of missing values and a large number of nominal levels and as such was removed from the feature set. The *initial diagnosis* which consisted of 63% missing values and other features that were not known at the time of admission into hospital were also discarded: *GDH, GCD, Discharge Destination, Date* and *Hour, Treatment*. The 14 attributes that remained were retrieved for the analysis.

After the features were selected, missing data values related to those features were replaced using the hotdeck method [110]. This substitutes a missing value by the value found in the most similar case. The work made use of the *rminer* package which uses a nearest neighbour algorithm which is applied for all of the features that contain values to find the closest example [111]. After the replacement of missing values within the data, several features were required to be transformed to facilitate modelling. With the premise of reducing the level of skew and improving symmetry of the underlying feature distribution, a logarithm transformation $y = ln(x + 1)$ was applied to both the *Previous Admissions* Feature and LoS features. This is a popular transformation that often improves regression results by smoothing right skewed variables.

**Table 6. A summary of Kalamata Greece dataset.**

| Kalamata Greece Dataset | | |
|---|---|---|
| **Feature** | **Feature type** | **Values** |
| *Age* | intervalised | $\{65 - 74, 75 - 84, 85+\}$ |
| *Gender* | binary | *Male, Female* |
| *Insurance Type* | categorical | *Private, Uninsured, Indigent, Other* |
| *Residence altitude* | intervalised | $\{0 - 100, 100 - 300, 300+ (m)\}$ |
| *Residence urbanity* | categorical | *Urban, semi-urban, rural* |
| *Residence Distance From Hospital* | discretised | $\{0 - 15, 15 - 30, 30 - 45, 45+ (km)\}$ |
| *Residence Medical Cover Type* | categorical | *Hospital, Regional Clinic, Rural* |
| *Day of Admission* | ordinal | *Days of the week* |
| *Month of Admission* | ordinal | *Months of the Year* |
| *ICD-10 codes* | categorical/ ordinal | *ICD-10 categories* |
| *Ward* | categorical | *Cardiology, Orthopaedic, General Surgery, Internal Medicine* |
| *Number of admissions* | continuous | 1, 2...100 |

Furthermore, any feature which contained an extremely large number of groupings were standardised to reduce the number of groupings, for example, *Main procedure* was reduced from hundreds of groupings to sixteen groupings and *Education* was transformed from 14 to 6 groupings. *Age* was also transformed so that it contained intervals.

## 4.3 Kalamata Greece dataset

The dataset consists of patients that were hospitalised in Kalamata general hospital between 2008 and 2012, a summary of the dataset can be found in Table 6. There were 4,403 patient records all aged over 65 years from both genders and varying diagnoses. Before a predictive model for the data could be built, data cleaning and preprocessing were carried out. This involved the removal of repeated records, irregular and irrelevant records as well as the manipulation of records that contain missing data. Additionally, any records that were found to have the same admission and discharge date were removed, (if a patient was admitted and discharged on the same day they were removed from the data). In terms of experimentation, a series of tests were performed to assess the performance of the most commonly used semi supervised learning (SSL) approaches: Self-training, Tri-training and Co-training. Each of the SSL approaches was then evaluated with several different standard classifier learners such as Naive Bayes, Decision Trees, PART, Multilayer Perceptron [107], the 3-Nearest-Neighbour algorithm, Sequential Minimal Optimisation (SMO) [108]. The analysis was performed using the WEKA data mining suite and the default WEKA parameter settings for each of the classifier learners were used to train the models. As mentioned previously, the accuracy of each of the algorithms was evaluated using stratified 10-fold cross validation. It was decided that four different proportions of labelled training data would be used to assess the influence of the amount of labelled data: 10%, 20%, 30% and 40%.

## 4.4 Characteristics of LoS data

Across many of the LoS studies in the literature there is commonality between the data types that are included despite the variation in application areas and approaches that have been used

to build models and to generate the results. In S1 Table, the key features that appear in multiple LoS studies are shown. *Age* is the feature that appears most frequently in all 15 studies that were compared, followed by *Gender* and *Primary Diagnosis*. *Age*, *Gender* and *Primary Diagnosis* are all features that are routinely collected when a patient is admitted to hospital and so often that information is readily available for analysis and doesn't need to be extracted. Other features such as *Previous Admissions* and *Comorbidity* require work to collect and although the information may be available at the hospital, it is either not recorded in a suitable format, or would take too much data collection effort to provide. These features in particular would likely require expert input to determine the comorbidities a patient has at the time of their admission and in the case of *Previous Admissions* further data collection effort would be needed to track down the previous records of admissions for a given patient. Features such as *Admission Type* are only required in cases where the overall LoS of patients from different wards or patient cohorts are needed to be compared. Otherwise, this feature is not required for the majority of studies that are concerned with specific patient conditions or cohorts. *Marital Status* is the only real outlier here as it is routinely collected when patients are admitted to hospital but does not appear to take precedence over the other features in this case and is only collected in four out of the fifteen studies. This is likely because *Marital Status* tends to be more important in elderly patient populations where a patient is more likely to live alone and be widowed or be married, and this disparity can influence LoS [119].

In the future, it may be beneficial for LoS studies to focus their efforts on collecting more routinely collected admission data as it would mean that LoS methodologies and models could be more directly comparable across hospital settings. Routinely collected data is explored in [120] where a novel approach for the prediction of LoS using only limited routinely collected data is presented. The data consisted of 6,543 patient admissions that were admitted to the Accident and Emergency department in Bronglais General Hospital, Aberystwyth, Wales, UK. The data consisted of 16 conditional features such as patient's *age*, *sex*, *first four characters of postcode* and *primary diagnosis* as well as up to twelve *secondary diagnoses*. The work classified *short stay* and *long stay* patients as a binary classification problem and made use of four traditional learning approaches and five fuzzy approaches. The best result of 76.39% was yielded using fuzzy-rough subset selection with a C4.5 decision tree based algorithm as the base learner. It was also determined that the *first four characters of postcode* was the most useful feature in classifying LoS which is significant given that the data is collected from a hospital in a rural setting, where the catchment areas are much larger and more demographically varied than that of urban hospitals.

## 5 Examination of state-of-the-art

In this section, A study which has recently been published and is at the forefront of LoS prediction for its respective patient cohort will be described in detail. The paper is entitled "Predicting in-patient length of stay after brain tumor surgery".

This paper presents a machine learning ensemble for predicting in-patient length of stay following craniotomy for brain tumor from preoperative patient features which are recorded in the National Inpatient Sample (NIS) [121]. The ensemble is then externally validated using the American College of Surgeons National Surgical Quality Improvement Program (NSQIP) database. 79,742,743 admissions registered in the NIS database from 2002–2011 were screened for inclusion in the training set and 1,195,376 admissions registered in the NSQIP database between 2012 and 2013 for inclusion in the external validation set. Eligibility for the study was determined via the international classification of diseases (ICD) 9 diagnosis codes for brain

tumors and procedure codes for craniotomy. This resulted in 41,222 admissions in the training set and 4,592 admissions in the validation set.

26 different features were selected for the study such as *age*, *race*, *sex* and specific *neurological diagnosis*. There is no discussion in the paper which detailed the process employed in order to decide which features were chosen. In terms of data preprocessing, any missing numerical data were imputed using the median value for the given variable, and a new binary variable was created that would denote the imputation. The motivation for doing this was that some algorithms are well suited to detect and leverage variable interactions, but being unable to function in the presence of missing data. Prior to training, 20% of the training set was randomly selected as the test set and excluded from training. The data that remained was then divided into five folds. Training was performed five times per algorithm, one for validation and four for training. The justification for choosing five fold cross-validation was the tradeoff between additional estimates of ensemble generalisability on cross-validation and algorithm training time. To offset the trade-off the model was evaluated against an entirely separate dataset. Parameters were tuned in accordance with each fold by creating a sub-fold training/validation split. The relative scores of each algorithm were evaluated using the root mean square logarithmic error (RMSLE) metric. The motivation for choosing this metric was that it penalises large errors less when both the predicted and actual LoS are small. The ensemble model was trained and validated in the same way as each individual algorithm and additional validation in the form of RMSLE being calculated for predictions made on the never before seen test set. The ensemble was then trained on 100% of the NIS database and externally validated with the NSQIP database to determine the LoS of patients after brain tumor surgery framed as a regression problem.

As previously mentioned, 29 classifier and regression-based learners were evaluated; these included tree-based models, support vector machines, neural networks, linear classifiers, Naive Bayes etc. The top performing learners were two gradient boosted trees and a Nystroem kernel support vector machine. The results of these experiments are not recorded in the paper and so the performance of the different algorithms cannot be compared. The best learners were combined with an elastic net [126] to create the ensemble model. The ensemble model recorded an RMSLE of 0.555 (95% CI,0.553–0.557) on internal validation, an RMSLE of 0.559 for the test set and an RMSLE of 0.631 for external validation. The paper also uses permutation importance to evaluate the relative importance of each feature [127] [128]. Using permutation importance constitutes the retraining of the ensemble with a version of the data in which all of the values for the feature in question are randomly permuted, which removes any predictive value of the feature but maintains its distribution. A comparison is then formed between the RMSLE of the original model and the mode is built with the permuted feature. In calculating the change in model performance for each permuted feature, the relative importance of each feature to the model can then be ranked with the most important features yielding the highest losses in model performance. The features that strongly influence LoS were found to be *nonelective craniotomy*, *preoperative pneumonia*, *preoperative sodium abnormality*, *preoperative weight loss* and *non-white race*. *Nonelective surgery* was found to independently increase predicted LoS from 6.3 to 9.7 days and *pneumonia* increased predicted LoS from 7.6 to 20.4 days. *Preoperative sodium abnormality*, *weight loss* and *non-white races* also significantly contributed to an increase in predicted LoS.

The work mentions its use of median imputation during the data preprocessing stage. Although using median imputation is more robust to the presence of outliers than mean imputation and is relatively simple to perform, the simplicity of this method is also its disadvantage. The overall distribution of the imputed features can become very distorted and the variance is underestimated as each of the missing values is replaced with an identical value [122]. It also

introduces bias as it disregards the relationship between features decreasing their correlation with each other [123].

Overall, this work offers a very robust solution to the prediction of LoS which could also be generalised to predict other medical outcomes. The study is robust because it heavily negates the effects of overfitting by ensuring that ensemble prediction are not tailored to the training dataset by using internal and external validation in the form of cross validation and test sets as well as an external validation set. In terms of limitations, the study highlights the potential to miss important LoS predictors because of the differences between the two datasets that were used. Hospital characteristics for example, were not encapsulated by the NSQIP database meaning that features such as *hospital location* that could have an influence on a patient's LoS are not considered.

## 6 Shortcomings of current LoS research

The management of hospital resources and the ability to model LoS is becoming an increasingly important issue. Healthcare systems are continually being developed in an attempt to model LoS in the hospital environment and this information can be used to improve the future allocation of resources. Whilst there have been various different approaches to modelling LOS, it remains a complex and uncertain area, often influenced by external and competing factors.

### 6.1 Towards a unified framework

Perhaps, the most apparent problem in the literature is that there is no single unified framework or approach for tackling the LoS prediction. This survey of the literature has demonstrated that almost all of the approaches that have been used to model LoS focus on datasets and (methodologies) that are disease, condition or patient-group specific. Allied to this, a large number of approaches are tuned according to domain knowledge of the hospital, the prediction technique that is employed or specific performance criteria in the data. It is challenging to ascertain whether some of these steps are justified or are simply included in order to achieve the "best" possible results. As mentioned previously, each approach to predicting LoS is only appropriate to the environment that it is being employed in. As such, the generated models are not suitable for adoption to more than one hospital setting. Ad-hoc methods result in the data preprocessing and model tuning steps being too specific to be used for the general case. The reason for this, is that the performance of a given approach will vary depending on a large number of competing factors such as the number of patients a hospital admits, a patient's diagnosis, the hospital's urban/rural location, particular procedures or processes in place and care units etc. It is this variation in collection and preprocessing of datasets that makes developing a general framework for LoS prediction a difficult task. Furthermore, many of the approaches that are employed are data specific which means that they are not comparable with one another. This work has shown that there is a large variety of techniques available for the prediction of hospital LoS. Notwithstanding the number of techniques that have been used, all of the models that have been created were based on a single institution or domain which leaves the generalisability of a model beyond its application in a local setting open to question. In order to address this concern, clearly there is a need for universal model or framework that takes the many differences that exist between different patient populations in terms of their demographics and social circumstances into account. Hence, there is a need for models which are trained using data that is routinely collected in the majority of hospitals. Using such data will directly enable the relative comparison of performance for LoS models in the literature as well as aid in making the models more applicable and generalisable to global healthcare settings particularly with the advent of electronic health records.

In order to better generalise the prediction of LoS so that it can be used in multiple health-care environments, it is necessary to develop a unified framework for modelling patient LoS. This unified framework would aim to ensure that the data collection, preprocessing and overall methodology are consistent and justified for each prediction approach. A unified framework that utilises a strict statistical analysis along with a data driven investigation will enable a more concrete examination of the factors that influence LoS in the general case. Each LoS prediction approach should follow a basic transparent process that encompasses the objectives and moti-vations, data collection, model generation and selection, evaluation, statistical testing as well as deployment and clinical acceptance. The LoS literature currently does not adhere to these basic principles and instead each study uses an ad-hoc methodology for prediction of LoS that is only applicable in a specific medical context. The use of a unified framework would ensure that every LoS approach is robust, fit for purpose and will enable the comparison of predictive performance between approaches. It is this level of robustness that will encourage the adoption of LoS prediction approaches by clinicians even if these approaches are not fully transparent to human scrutiny. Similarly, S1 Table demonstrated that the data features that are collected in the literature have some commonalities despite the differences in application to specific patient groups or diseases. A unified framework could specify or inform the types of data and features to be used in a given LoS study, these could include the key features that appear in a large num-ber of LoS studies and are thus considered to have an influence on the LoS of a patient. The fact that *Age*, *Gender* and *Primary Diagnosis* appear most frequently clearly indicates their per-ceived influence on LoS. If the work is centred around a specific condition, it could include the general features that have a perceived influence on LoS as well as the features that are impor-tant in determining LoS for that particular condition. An example of such a framework is shown in Fig 5.

## 6.2 Lack of rural features and nursing admission data

Many of the approaches that have been used to model LoS utilise data in the form of diagnostic electronic records that are either prospectively and/or retrospectively gathered within 24 hours of admission. This data typically takes into account a physician's diagnosis of a patient's pre-senting complaint. As it currently stands, there is no work in the literature which considers the influence of nursing or other medical staff admission data on the prediction of LoS. This could offer a unique standpoint on the problem of assessing LoS as it has been noted that nurses spend far more time with patients than physicians [112] [113]. In addition to this, nursing admission data takes a wide range of different factors that could contribute to an extended LoS in hospital into consideration. This type of data largely focuses on a patients social background and includes data such as a patient's home situation, lifestyle habits and whether or not they have a carer prepared to tend to them. This is important, as patient LoS can be prolonged by a patient being perfectly healthy but not having a suitable residence to return to after their stay. This particular case is prevalent in older patients where a carers burden is likely to be greater [114]. It also encapsulates many in-hospital assessments that are carried out by nurses on a daily basis such as a patient's level of frailty and their willingness to comply with instructions.

Equally, the majority of studies in the literature centre on hospitals that are located within an urban and/or densely populated area. A hospital's urbanity could play an important part in determining a patient's predicted LoS. This is because patient's may face logistical challenges when attempting to gain access to rural healthcare. Rural hospitals tend to have large catch-ment areas with diverse demographics which means that a hospital could be located a signifi-cant distance from a patient's home; making access to ambulatory care and services difficult [115]. Given these challenges it is possible that a patient may seek medical care from a general

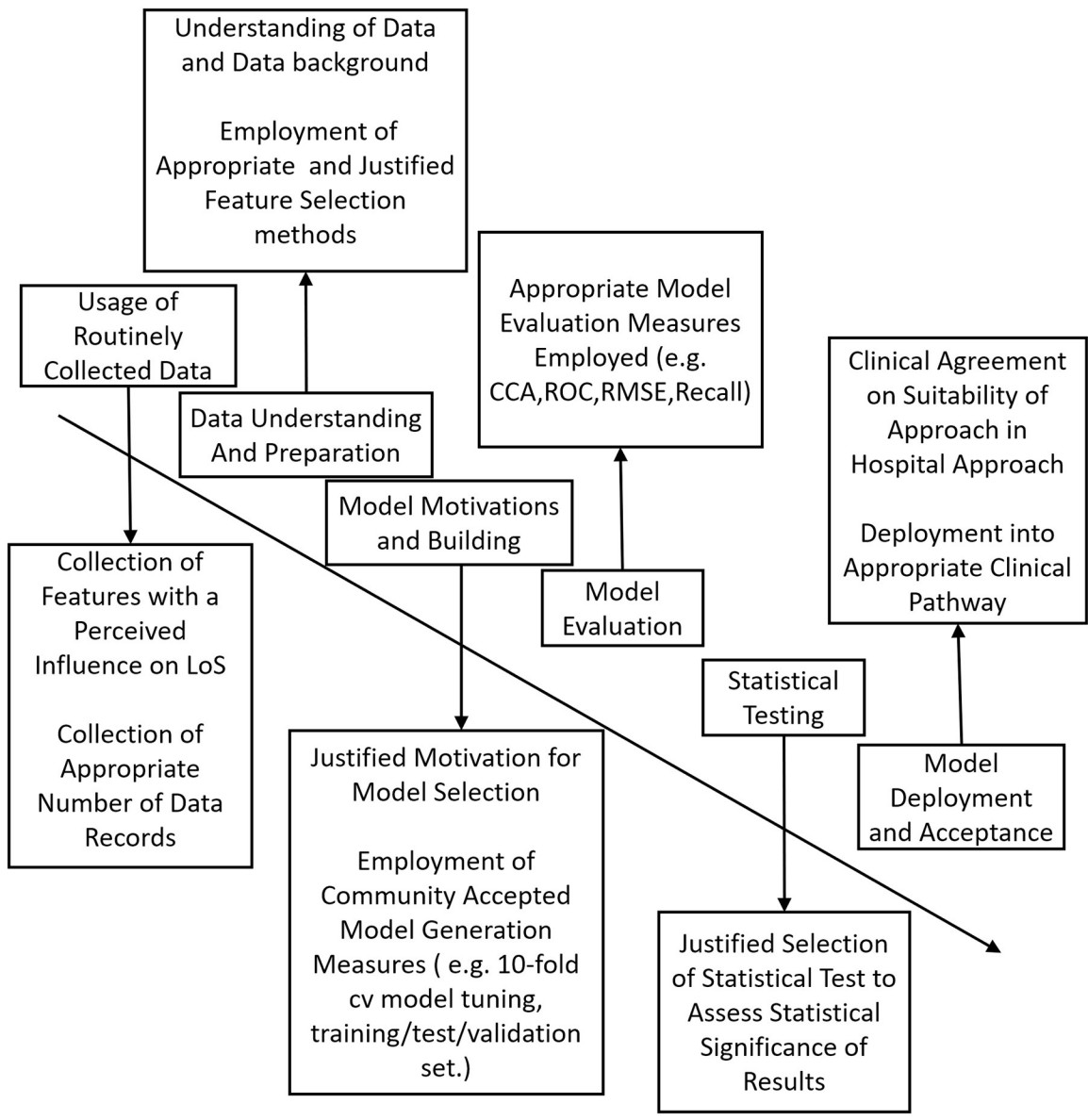

**Fig 5. Unified framework for LoS prediction.**

practitioner or emergency services that are closer to home, this is particularly prevalent in end-of-life care [116]. Further to this, patients that have been admitted to a rural hospital may have a prolonged LoS because of limited availability of transport or services to support them in the community [117]. More specifically, the work in [118] showed that although 17% of the U. S population lived in rural areas in 2010, only 12% of total hospitalisations, (11% of the days of care and 6% of the overall in-patient procedures) were provided at rural hospitals. The work revealed that patients hospitalised in urban hospitals were more likely to have procedures performed during their hospitalisation and were more likely to have a greater number of procedures performed than their rural counterparts. It was inferred that this could be due to shortages of speciality physicians in remote rural areas, lack of staff that were skilled in surgery

or the absence of costly equipment required for surgical procedures. The urbanity or rurality of a hospital environment is therefore an important factor in the assessment of patient LoS.

The lack of rural features and nursing admission data may be remedied with the use of routinely collected data. Currently, there is a push to digitise health data across the world in the hope that it is possible to more easily improve the level of patient care. The amount of routinely collected patient healthcare data that is being accumulated and stored is vast and rapidly expanding [129]. There are many advantages that like all other types of data, electronic health records could yield over cohort based studies. By employing electronic health records, one is able to observe considerably more metrics, at more time points and at less cost than prospective cohort based studies [130]. Moreover, the same set of data can be used to predict a wide range of different clinical outcomes as opposed to cohort studies which do not allow this. Patient populations that are derived from electronic health records tend to be more reflective of the real-world because they are not reliant on volunteer participation and can cover a much wider patient demographic. The use of electronic health records can incorporate free text in its prediction of LoS using free-text notes from doctors and nurses. This use of free-text analysis could enable a new level of explainability for LoS predictions. Although clinicians have historically distrusted ANNs because of their opaqueness, as previously mentioned there have been studies which have investigated the interpretation of such models [131] [132]. These approaches may address concerns that black-box methods such as NNs are untrustworthy as they enable a clinician to view what data the model "looked at" for each individual patient. A clinician can then determine if the given prediction was credible.

## 6.3 Adoption of machine learning in healthcare

Lastly, with the use of machine learning and artificial intelligence in healthcare becoming ubiquitous, there is an increased interest in approaches that are interpretable and can be trusted by clinicians and hospital managers. The use of black-box approaches in the prediction of LoS has led to some hesitation in the deployment of such approaches because the cost of model misclassification is so high and black-box approaches have no intepretability. The lack of interpretability in machine learning approaches can have potentially life threatening consequences. For example in [124] [125] work on building a classification model for labelling pneumonia as high or low risk of in-hospital death was performed. An ANN learner was shown to be the best performing classifier. Investigation into this problem with the use of regression models revealed that one of the top predicting features was *patient history of asthma*. The NN was predicting that patient's who had a history of asthma has a lower of risk of in-hospital death from pneumonia. In reality, this prediction was counter intuitive as patients with asthma are at higher risk of serious health complications including death. The asthma patients in this case were provided timely interventions of a higher acuity than patients without asthma meaning that they were actually more likely to survive. As machine learning becomes increasingly prominent in a healthcare setting, interpretability and transparency of machine learning techniques will become more important. The majority of currently employed machine learning LoS prediction techniques in healthcare solely provide predictions but in practice reliable reasoning will be needed to convince medical clinicians to use such approaches.

Furthermore, alternative modelling approaches to predicting LoS may also encourage the use of machine learning and data mining in healthcare. Generally, the dynamics that govern a hospital environment, the flow of patients, and their respective LoS require models that are good at dealing with the uncertainty, variability and complexity within the healthcare setting. As previously mentioned, there are many different types of uncertainty that are present within LoS data. To better understand the underlying relationships in the data and how they may be

connected, effective and robust methods for modelling uncertainty are therefore essential. There are currently a wide variety of approaches that can model uncertainty and complexity which are underused in the context of LoS. ANNs and SVMs can be useful for building classification models with good predictive accuracy and can discover complex non-linear relationships in the data. However, they cannot derive models that are transparent to human scrutiny without additional research effort into model interpretation, and between them this interpretability is limited. Conversely, there are methods which can build models that are humanly transparent and thus will enable the modelling of different types of uncertainty that are present in a healthcare environment. For instance, probability-based modelling is perhaps the most well-known and employed in the area of machine learning and is used in techniques such as Bayesian belief networks mentioned previously in this report. These approaches model the probabilistic uncertainty of events. Other possibilistic approaches also exist such as Fuzzy Systems [133]. These types of approaches can be very powerful for modelling linguistic concepts of partial membership or 'vagueness' that are natural to humans, things such as the concept of 'tall' or 'hot' for example which are essentially human concepts [134]. However, fuzzy systems use these concepts very successfully in order to model real-world relationships [135] which makes them an ideal candidate to be used in the prediction of LoS. These fuzzy inference systems could also function as an alternative to Markov models with state probabilities being represented in the form of membership functions and a set of fuzzy rules which govern how the inference system makes decisions about a patient's LoS. Presently, there is limited literature which makes use of fuzzy inference systems and techniques to tackle the LoS prediction problem.

## 7 Conclusion

The ability to predict LoS can provide a clinical indicator of the health status of a patient as well as assist in predicting the level of care that is required. It also aids hospital staff with improved prediction of bed and ward utilisation. LoS varies with respect to many factors including severity of illness, diagnosis and a variety of patient factors. This paper provides a review of LoS prediction methods, their respective shortcomings as well as the types of data and features that have been used in the literature. Despite the continuing efforts to predict and reduce the LoS of patients, current research in this domain remains ad-hoc; as such, the model tuning and data preprocessing steps are too specific and result in a large proportion of the current prediction mechanisms being restricted to the hospital that they were employed in. Additionally, several studies focus on hospitals which are contained within very densely populated areas.

Adopting a unified framework for the prediction of LoS could yield a more reliable estimate of the LoS; this can be used across different hospitals where patient populations will be homogeneous with similar demographics and comorbidities amongst patients. Expanding the influence of the models that are generated as part of a unified framework would ensure that the prediction approaches in place are suitably robust as the datasets used would be considerably larger. Increasing the size and coverage of these datasets would improve the ability to detect complex patterns in LoS which could lead to a reduction in prolonged patient LoS where patients have a higher risk of exposure to adverse effects such as hospital acquired infections. Ultimately, the inherent complexity and uncertainty of healthcare systems combined with the vast amounts of electronic healthcare data currently being collected, necessitates prediction methods which are broadly applicable and are capable of modelling uncertainty.

Further research is required to explore novel methods such as fuzzy systems which could build upon the success of current models as well as further exploration of black-box approaches

and model interpretability. This is important, as in general, healthcare workers are overwhelmed by the sheer number of patients that they are required to care for, the associated tasks required of them and the amount of data generated by the patients. Machine learning implementations and their explanations, if not sufficiently interpretable, could further hamper the day-to-day effort, of a healthcare worker. Balancing the interpretability of such models with the overall prediction performance that they provide will be a key challenge in the future of LoS prediction.

## Supporting information

**S1 Table. Feature commonalities across 15 LoS studies.**
(TIFF)

**S1 Prisma Checklist. Prisma checklist.**
(PDF)

## Author Contributions

**Conceptualization:** Reyer Zwiggelaar, Neil Mac Parthaláin.

**Data curation:** Kieran Stone.

**Formal analysis:** Kieran Stone.

**Funding acquisition:** Reyer Zwiggelaar, Phil Jones, Neil Mac Parthaláin.

**Investigation:** Kieran Stone.

**Methodology:** Kieran Stone.

**Project administration:** Kieran Stone.

**Resources:** Kieran Stone.

**Software:** Kieran Stone.

**Supervision:** Reyer Zwiggelaar, Phil Jones, Neil Mac Parthaláin.

**Validation:** Kieran Stone.

**Visualization:** Kieran Stone.

**Writing – original draft:** Kieran Stone.

**Writing – review & editing:** Kieran Stone, Reyer Zwiggelaar.

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
