## [Decision Letter · Decision Letter 0]

19 Aug 2021

PDIG-D-21-00027The Prediction of Hospital Length of Stay: Towards a Unified FrameworkPLOS Digital Health

Dear Dr. Stone,

Thank you for submitting your manuscript to PLOS Digital Health. After careful consideration, we feel that it has merit but does not fully meet PLOS Digital Health’s publication criteria as it currently stands. Therefore, we invite you to submit a revised version of the manuscript that addresses the points raised during the review process.

Refer to each change using line numbers throughout, please.

We look forward to receiving your revised manuscript.

Kind regards,

Martin G Frasch

Section Editor

PLOS Digital Health

Journal Requirements:

1. We note that your submission is a Systematic or Scoping Review, which should be uploaded as a 'Research Article' article type in Editorial Manager. Please change the article type to 'Research Article', and ensure all submission questions are completed for this article type. Please note that, if accepted, Research Articles have an APC associated with publication.

2. Please ensure that the CRediT author contributions listed for every co-author are completed accurately and in full. 

At this stage, the following Authors require contributions: Kieran Stone, Reyer Zwiggelaar, Phil Jones, and Neil Mac Parthalain.  Please ensure that the full contributions of each author are acknowledged in the "Add/Edit/Remove Authors" section of our submission form.

3. We ask that a manuscript source file is provided at Revision. Please upload your manuscript file as a .doc, .docx, .rtf or .tex. If you are providing a .tex file, please upload it under the item type ‘LaTeX Source File’ and leave your .pdf version as the item type 'Manuscript'.

4. Please provide separate figure files in .tif or .eps format only, and remove any figures embedded in your manuscript file. If you are using LaTeX, you do not need to remove embedded figures.

5. Please provide a complete Data Availability Statement in the submission form, ensuring you include all necessary access information or a reason for why you are unable to make your data freely accessible. Note that it is not acceptable for the authors to be the sole named individuals responsible for ensuring data access.

PLOS defines a study's minimal data set as the underlying data used to reach the conclusions drawn in the manuscript and any additional data required to replicate the reported study findings in their entirety. Any potentially identifying patient information must be fully anonymized. 

If your research concerns only data provided within your submission, please write "All data are in the manuscript and/or supporting information files" as your Data Availability Statement.

6. Please amend your detailed Financial Disclosure statement. This is published with the article, therefore should be completed in full sentences and contain the exact wording you wish to be published.

i). State what role the funders took in the study. If the funders had no role in your study, please state: "The funders had no role in study design, data collection and analysis, decision to publish, or preparation of the manuscript."

ii). If any authors received a salary from any of your funders, please state which authors and which funders.

If you did not receive any funding for this study, please simply state: "The authors received no specific funding for this work."

Additional Editor Comments (if provided):

Reviewers' comments:

Reviewer's Responses to Questions

**Comments to the Author**

1. Does this manuscript meet PLOS Digital Health’s publication criteria? Is the manuscript technically sound, and do the data support the conclusions? The manuscript must describe methodologically and ethically rigorous research with conclusions that are appropriately drawn based on the data presented.

Reviewer #1: No

Reviewer #2: Yes

Reviewer #3: Yes

2. Has the statistical analysis been performed appropriately and rigorously?

Reviewer #1: N/A

Reviewer #2: Yes

Reviewer #3: N/A

3. Have the authors made all data underlying the findings in their manuscript fully available (please refer to the Data Availability Statement at the start of the manuscript PDF file)?

Reviewer #1: No

Reviewer #2: Yes

Reviewer #3: Yes

4. Is the manuscript presented in an intelligible fashion and written in standard English?

Reviewer #1: Yes

Reviewer #2: Yes

Reviewer #3: Yes

5. Review Comments to the Author

Reviewer #1: This manuscript presents an overview of methods and datasets for modelling or forecasting in-patients hospital length of stay.

The abstract and introduction present this paper as a systematic review (or rapid review). However, the structure and content of the paper do not match a systematic review usual structure, despite the mention of the PRISMA checklist. For example, the methods are briefly described as a sub-section of the introduction (not in a separate section), and there is no ‘findings’ section as such, separate from discussion.

The methods subsection misses some essential information about the review process. First of all, it mentions the search for (and inclusion of) ‘surveys’ – from which papers are retrieved, and then studies included. This is very confusing. It is not known what ‘surveys’ mean in this context. Are they systematic reviews? Is this a systematic review of systematic reviews?

It also does not describe the review process: e.g. inclusion/exclusion criteria; the search strategy; criteria for screening results; methods for data extraction; how many researchers involved in the screening (reliability); methods for analysis… It is impossible to assess the quality of this study without this information.

The abstract and introduction also suggest that the analysis is done with a unifying framework. However this is not the case – (the need for) a framework is only presented, it seems, as a conclusion.

The rest of the paper summarises models and datasets for LoS, but we have no idea about the sources of this information – they are not presented as findings from the review, more as a discussion.

Overall this paper feels as materials written for, or derived from, another document, such as a thesis, book chapter or project report. It also makes reference to an Appendix which is not provided in this submission (I could not find it).

I suggest the authors rewrite this paper in the structure of a systematic review. The topic is potentially interesting and worth publishing.

Reviewer #2: This manuscript is a systematic review paper to predict the length of stay (LOS) in hospitals.

This manuscript investigated papers from 1970 to 2019, and clearly described various prediction techniques, datasets in research, and experiments in the research. In addition, as a review paper, enough references have been investigated, and the descriptions of each technique are written in detail so that they can be easily read.

However, this manuscript has the following weaknesses.

1. Since this manuscript is a review paper, it would be nice to have a table organized as a whole so that it is easy to find each reference. In other words, a classification table such as category for the final extracted 93 studies is required (similar to Table 5). For example, I recommend adding a table matching the reference with the taxonomy created in Figure 2.

2. The order of the subsection of Chapter 2 is 2.1 Operational research based approaches, 2.2 Statistical and Arithmetic approaches, 2.3 Machine Learning and Data Mining approaches. Therefore, the taxonomy in Figure 2 should also be arranged in the order of subsections. Currently, the name and order are different.

3. Sections 2.3, 2.3.1, and 2.3.2 do not match the name and structure of Figure 2. Therefore, either modify the machine learning part of the taxonomy or modify this section according to the structure of the taxonomy.

4. Chapter 4 has only one section in section 4.1, so either add content to make it after section 4.2 or remove the title of section 4.1 and combine it into section 4.

5. In this manuscript, the papers up to 2019 were well organized, but it seems that studies using more diverse techniques for machine learning were conducted in 2020 and 2021. I think that, at least, machine learning techniques using the latest deep learning need to be added.

Reviewer #3: Thank you for the opportunity to review this comprehensive literature review conducted using a rapid evidence assessment methodology. I appreciated the approach in results with the focus on approaches to modeling with covers a wide range of methodological approaches. I also appreciated the in depth case/exemplar that the authors took the readers through. I also appreciated considerations regarding implementation and interpretability.

A couple of recommendations: (1) the ethical implications of modeling LOS are not adequately covered (i.e., in use could a health system use this type of modeling strategy to deny care to those perceived to be high utilizers?) and (2) the inherent opportunities for bias using EHR data in terms of modeling approaches. We know that clinicians have inherent biases so tests ordered, assessments documented, will also carry the inherent bias of these processes and will be included in these models.

Overall, I found this to be a comprehensive literature review and important contribution.

6. PLOS authors have the option to publish the peer review history of their article (what does this mean?). If published, this will include your full peer review and any attached files.

**Do you want your identity to be public for this peer review?** For information about this choice, including consent withdrawal, please see our Privacy Policy.

Reviewer #1: No

Reviewer #2: No

Reviewer #3: No

---

## [Decision Letter · Decision Letter 1]

6 Jan 2022

The Prediction of Hospital Length of Stay: Towards a Unified Framework

PDIG-D-21-00027R1

Dear Dr. Stone,

We're pleased to inform you that your manuscript has been judged scientifically suitable for publication and will be formally accepted for publication once it meets all outstanding technical requirements. 

Within one week, you'll receive an e-mail detailing the required amendments. When these have been addressed, you'll receive a formal acceptance letter and your manuscript will be scheduled for publication. The journal will begin publishing content in early 2022.

An invoice for payment will follow shortly after the formal acceptance. To ensure an efficient process, please log into Editorial Manager at https://www.editorialmanager.com/pdig/ click the 'Update My Information' link at the top of the page, and double check that your user information is up-to-date. If you have any billing related questions, please contact our Author Billing department directly at authorbilling@plos.org.

Kind regards,

Dukyong Yoon

Academic Editor

PLOS Digital Health

Reviewers' comments:

Reviewer's Responses to Questions

**Comments to the Author**

1. If the authors have adequately addressed your comments raised in a previous round of review and you feel that this manuscript is now acceptable for publication, you may indicate that here to bypass the “Comments to the Author” section, enter your conflict of interest statement in the “Confidential to Editor” section, and submit your "Accept" recommendation.

Reviewer #2: All comments have been addressed

2. Does this manuscript meet PLOS Digital Health’s publication criteria? Is the manuscript technically sound, and do the data support the conclusions? The manuscript must describe methodologically and ethically rigorous research with conclusions that are appropriately drawn based on the data presented.

Reviewer #2: Yes

3. Has the statistical analysis been performed appropriately and rigorously?

Reviewer #2: N/A

4. Have the authors made all data underlying the findings in their manuscript fully available (please refer to the Data Availability Statement at the start of the manuscript PDF file)?

Reviewer #2: Yes

5. Is the manuscript presented in an intelligible fashion and written in standard English?

Reviewer #2: Yes

6. Review Comments to the Author

Reviewer #2: This manuscript seems to reflect the reviewers' recommendations well. Therefore, I accept this manuscript.

7. PLOS authors have the option to publish the peer review history of their article (what does this mean?). If published, this will include your full peer review and any attached files.

**Do you want your identity to be public for this peer review?** For information about this choice, including consent withdrawal, please see our Privacy Policy.

Reviewer #2: No
